# A New Control Algorithm to Increase the Stability of Wind–Hydro Power Plants in Isolated Systems: El Hierro as a Case Study

Agustín Marrero [1], Jaime González [1], José A. Carta [2] and Pedro Cabrera [2,*]

1   Department of Electronics and Automatics Engineering, Universidad de Las Palmas de Gran Canaria, Campus de Tafira s/n, 35017 Las Palmas de Gran Canaria, Spain
2   Department of Mechanical Engineering, Universidad de Las Palmas de Gran Canaria, Campus de Tafira s/n, 35017 Las Palmas de Gran Canaria, Spain
*   Correspondence: pedro.cabrerasantana@ulpgc.es; Tel.: +34-928-45-9887

**Abstract:** The present paper proposes the implementation of a new algorithm for the control of the speed regulators of Pelton wheel turbines, used in many of the pumped hydroelectric energy storage systems that operate in isolated electrical systems with high renewable energy participation. This algorithm differs substantially from the standard developments which use PID or PI governors in that, in addition to acting on the nozzle needles and deflectors, it incorporates a new inner-loop pressure stabilization circuit to improve frequency regulation and dampen the effects of the pressure waves that are generated when regulating needle position. The proposed algorithm has been implemented in the Gorona del Viento wind–hydro power plant, an installation which supplies the primary energy needs of the island of El Hierro (Canary Islands, Spain). Although, as well as its wind and hydro generation systems, the plant also has a diesel engine based generation system, the validation of the results of the study presented here focuses on situations in which frequency control is provided exclusively by the hydroelectric plant. It is shown that implementation of the proposed algorithm, which replaces the previous control system based on a classical PI governor, is able to damp the pressure wave that originates in the long penstock of the plant in the face of variations in non-dispatchable renewable generation, a situation which occurred with a high degree of relative frequency in the case study. The damper has enabled a substantial reduction in the cumulative time and the number of times that frequency exceeded different safety margins. Damper incorporation also reduced the number of under-frequency pump unit load shedding events by 93%.

**Keywords:** isolated systems; pumped energy storage; hydro-turbine control; quality of electrical energy; wind power

## 1. Introduction

In the fight against climate change, the European Union has drawn up the Energy Roadmap 2050 with a view to substantially reducing the emission of pollutants into the atmosphere [1]. The energy generation sector will play a vital role in facilitating a profitable transition to a low carbon emission economy. Among other measures, this will entail replacing fossil fuels with non-polluting renewable energy sources. As indicated by Ahshan et al. [2], wind and solar energy have become the best alternative sources of clean and sustainable energy. According to Caralis et al. [3], wind energy exploitation has grown rapidly worldwide and is the most competitive renewable energy source from a commercial and economic point of view. According to these authors [3], several power systems in the world are supplied and will be supplied in the near future by large fractions of wind energy. However, given the temporal variability of most such energy sources and the uncertainty associated with them, their widespread use implies the generation of a series of impacts, both economic and technical, for the electricity supply industry. According to

Ercan and Kentel [4], one of the main challenges in the use of wind energy is mitigating its strongly intermittent and fluctuating behaviour, which has negative effects on grid safety and stability. The technical impacts can affect the correct functioning of the electrical system and the equipment connected to it, as well as the quality of the energy supplied [5,6]. One strategy to mitigate impacts of a technical nature that result from implementation of renewable energy sources involves using energy storage systems. These include pumped hydroelectric energy storage (PHES), considered the most mature technology used in high power applications [7,8]. According to Ercan and Kentel [4], PHES is currently the most viable form of large-scale energy storage, and the operation of renewable energy systems together with pumped storage hydropower plants is highly efficient. As reported by Javaid et al. [9], hydro power and solar and wind energy constitute the main renewable sources of energy. According to Rehman et al. [10], the storage flexibility and capacity of hydroelectric energy enable grid stability improvements and facilitate the deployment of other intermittent renewable energy sources, such as wind and solar. According to these authors [10], it can be concluded from the technical review that they performed in relation to PHES that this is the most suitable technology for small autonomous island electrical systems and massive energy storage, with PHES energy efficiency ranging in practice between 70% and 80%. They also indicate that PHES based on photovoltaic energy has only been used on a very small scale. As a result of all the above, a renewed interest is emerging worldwide in the use of PHES along with demands for the rehabilitation of old small hydropower plants [10]. According to Javed et al. [11], a large number of studies in the literature recommend high levels of wind penetration in order to energize remote areas and the use of PHES as energy storage to fill and decrease the energy gaps caused by variations in wind availability. According to these authors [11], in the bibliographic review they undertook they observed that PHES systems that use wind energy are technically and economically viable for different geographic locations throughout the world, which they cite in their work. However, as reported by Ali et al. [12] in the systematic literature review they carried out, there exist various barriers to the deployment of PHES applications. However, as pointed out by Lei [13], some regions pose obstacles for the development and use of hydroelectric energy given the characteristics of this energy source, promoting the construction of small- and large-scale hydroelectric plants is fundamental. In fact, Ma et al. [14] highlight that hydroelectric energy has turned out to be an important guarantee for social and economic evolution. As reported by Yang et al. [15], as energy systems become more and more dependent on a continually growing combination of intermittent renewable energies, hydroelectric energy units are being required to provide increasingly aggressive frequency control. Thus, in small electrical power systems, like those installed in the Canary Islands (Spain), PHES systems can play a key role as a way to increase renewable energy exploitation [16–20]. Some PHES systems are already in operation [21–23], while others are in the construction stage [24]. However, according to [10], PHES systems require, in addition to advances in turbine design to improve plant efficiency and flexibility, new strategies to optimize storage capacity and maximize the rate of return of the plant. Safe and stable operation of hydropower plants must be guaranteed to achieve clean energy production and renewable integration in the system [25].

However, the implementation of PHES that gives rise to a rapid response in such types of system can provoke pressure variations in the hydraulic circuit when the PHES control system attempts to mitigate the power variations generated by the non-dispatchable renewable energy sources. As pointed out by Ma et al. [14], the hydroelectric turbine is the key energy device in hydroelectric plants. The problems related to turbine regulation with large load variations in the electrical system are as yet unresolved and continue to pose challenges to the control community [26].

Considering the two most commonly used hydro turbine classification types based on the water pressure change when passing through the rotor [27], impulse turbines are particularly suited to high head applications. In Pelton turbines, water is piped at high

pressure to one or more nozzles where it expands completely to atmospheric pressure (a simple hydroelectric scheme is shown in Figure 1).

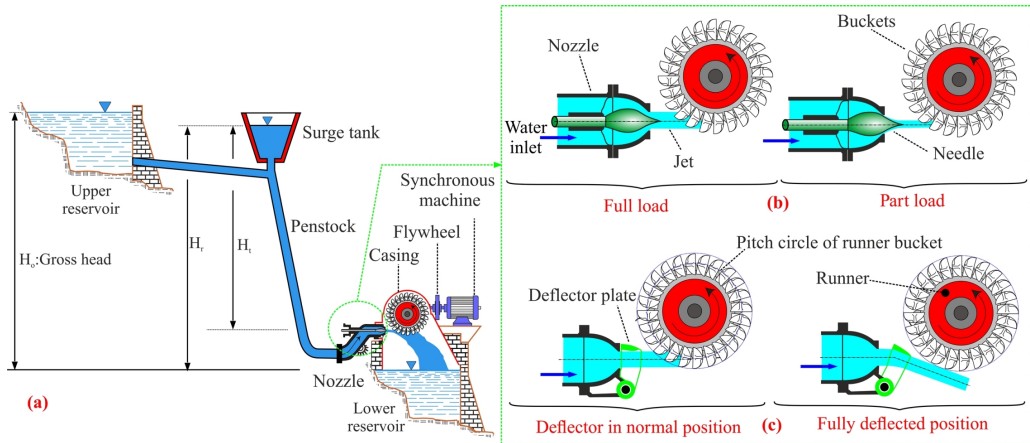

**Figure 1.** (**a**) Hydroelectric layout of a Pelton turbine. (**b**) Regulation with a needle valve. (**c**) Regulation with a deflector plate.

The jet that emerges from each nozzle hits the blades (or buckets) which are spaced around the periphery of a circular disc which constitutes the rotor [28]. This rotor is generally coupled to an electric generator, which has to work at synchronous speed, to produce the torque and power that is demanded [28]. Control of the mechanical power generated by Pelton turbines to ensure the frequency closely matches the grid values can be carried out as follows [28]: (i) regulating the flow rate of water emitted by the nozzles through the use of a spear (or needle) valve, controlled by means of servomotors, which is axially displaced within the nozzle to vary the diameter of the jet (Figure 1b); (ii) regulating the deflector plates, by means of servomotors, to divert part of the water emitted by the nozzle in such a way that only a certain percentage of the jet falls on the runner bucket (Figure 1c); (iii) coordinated regulation of both needle and deflector plate.

A surge tank (or surge chamber) is installed in some systems to absorb and dissipate part of the pressure fluctuations generated when the needle valves are closed rapidly (Figure 1) [28].

### 1.1. Literature Review on Control to Increase Wind–Hydro Power Plant Stability in Isolated Systems

Xu et al. [29], in a recent review of the literature on dynamic models and stability analysis approaches for a hydro-turbine governing system, indicate the main challenges that need to be tackled. These include [29] the control methods of hydro-turbine governing systems with intermittent renewable energies. In this context, it should be noted that diverse control alternatives have been proposed to solve the problems that arise due to the presence of external perturbations generated by variations of non-dispatched renewable energies [29]. These range from modifications of the widely used classical or traditional control systems [30] to new control model proposals based on the use of artificial intelligence (AI) techniques [31] or multi-objective optimization integrated systems [32]. The classical proportional-integral-derivative (PID) controller is the most commonly used type in the governor [26], although it has some drawbacks when dealing with complex systems [29]. As reported by [33], the criteria commonly used to adjust PID governors would not be appropriate in hydroelectric plants with long penstocks. These authors [33] propose using a second-order proportional integral (PI) control system for the turbine-penstock based on a grouped parameter approach. Various criteria have been proposed in the literature to adjust the parameters of traditional governors. Fang et al. [34] proposed using an improved particle swarm optimization algorithm to adjust PID gains. Jiang et al. [35] proposed the use of evolutionary programming to optimize the PID parameters and highlighted the advantage of the fast response time of their proposal. Among the AI techniques that have

been put forward, the use of non-linear model predictive control (MPC) strategies in the works of Mennemann et al. [36] and Reigstad and Uhlen [37] can be highlighted. Reigstad and Uhlen [37] argue that MPC is a well developed and extensively used method in process control that offers major advantages in comparison with traditional PID controllers. However, the same authors [37] stress that the controllers should be implemented and properly tested in a laboratory to better verify their performance.

### 1.2. Aim, Novelty and Key Contributions of This Paper

The literature analysis revealed that diverse models have been proposed to control the frequency of isolated electrical systems with high renewable energy participation and which use PHES as a frequency regulation system. The results obtained with these models have been analysed but not validated with test data.

The aim of this paper is to propose a new algorithm to manage the speed regulators of impulse-type hydro turbines, such as the Pelton wheel turbine, which are integrated in many of these systems.

The novelty of this paper lies in the fact that this approach differs substantially from the standard approaches which use PID or PI. The proposed governor system has a new inner-loop pressure stabilization circuit to improve frequency regulations and mitigate the effects of the pressure waves generated when regulating needle position. The aim with this new strategy is to optimize turbine response time in the face of variations in non-dispatchable renewable energy generation, when frequency control is exclusively provided by the PHES, and hence minimize load shedding and the associated frequency variations.

The vast majority of the interesting models proposed for isolated electrical systems with high renewable participation [38–41] have been developed in a theoretical context. However, in this work, theoretically and experimentally obtained results are used to validate test data obtained from the Gorona del Viento wind–hydro power plant installed in El Hierro island. After replacing the previous algorithms based solely on a classical PI governor with the proposed algorithm, this validation is performed in situations in which the thermal plant is disconnected, the energy obtained from the wind farm (WF) and the PHES system is sufficient to cover demand, and the frequency control is provided exclusively by the wind–hydro plant. This implementation, with its corresponding validation in a real PHES system, has not been detected in the current scientific literature. Such situations can arise with relative frequency in the system of the case study and, as indicated by Wang et al. [42], must be analysed to ensure the safe and stable operation of the electrical system.

## 2. Modelling of the System

In this section, a general overview is first provided of the system under consideration and the initial assumptions used in its modelling. The subroutines that make up the proposed algorithm are then described and the subroutines developed.

### 2.1. General Configuration of the System

Figure 2 shows a representative outline of the general configuration of the system. In it can be seen the renewable generation system, in this case comprising a wind farm, and the conventional generation system, in this case a thermal power station. The energy generation technologies and the different sectors (domestic, industrial and commercial) which it is intended to supply the energy demanded at each instant to are situated left of the vertical line which represents the utility grid. Situated to the right of the same vertical line is the representation of the PHES system (the energy flows are indicated by green-coloured arrows). The PHES system has a set of n Pelton turbines, which receive the feed water from the penstock manifold (the water flows are indicated by blue arrows), transforming the water pressure energy into mechanical rotational energy. The hydro turbines are connected to synchronous generators, with automatic voltage regulation, that are mechanically coupled to flywheels. Each unit has a speed regulator—which will be governed by the proposed control algorithm—that acts on its needles and deflectors,

and a step-up transformer for the grid connection of each group. The PHES system has two reservoirs, an upper one responsible for supplying water to the penstock manifold via a long penstock, and a lower one in which the water emerging from the hydraulic turbines is stored.

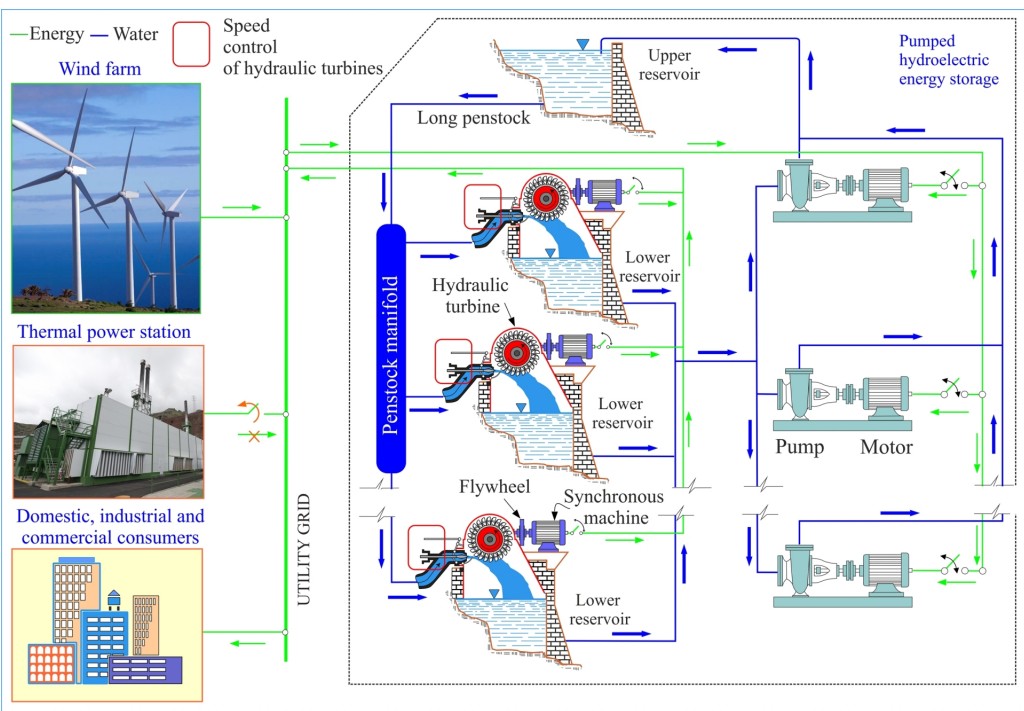

**Figure 2.** General outline of the configuration of the system.

The water stored in the lower reservoir is taken to the upper reservoir by a set of impulsion pumps which are connected to electric motors which receive the energy required for their operation from the utility grid.

The most important initial assumptions used in the modelling of the system are given below.

(a)　Although a thermal plant is one of the components of the system shown in Figure 2, the situation considered in the analysis that is undertaken is with the thermal plant disconnected and the energy from the WF and PHES sufficient to cover demand.

(b)　In the analysed scenarios, both the pumps which pipe water from the lower to upper reservoir and the loads external to the generation system are considered constant loads in the transient period studied.

(c)　System frequency control is provided exclusively by the PHES system.

(d)　Wind generation is modelled as a dynamic load to which real records of wind power variations can be incorporated, and with respect to which ramps or steps resulting from that generation can be simulated.

### 2.2. Basic Outline of the Proposed Algorithm

Figure 3 shows a basic outline of the proposed control algorithm, including the connections, from the control point of view, of the different elements that make up the system. The left-hand side column of tan-coloured boxes in Figure 3 shows the governor elements (regulation with a needle valve and regulation with a deflector plate) of the n hydraulic turbines considered. Each governor-*i* has three input signals: the turbine rotating speed ($\omega$) which the power system model provides (represented by the yellow-coloured box in the upper right-hand part of Figure 3), an error signal (*epi*) generated by the control algorithm proposed in this paper (represented by the cyan-coloured box in the upper left-hand part of Figure 3), and the operating reference power of turbine-*i*

which the automatic generation control (AGC) generates. The latter is represented by the violet-coloured box situated in the upper right-hand part of Figure 3.

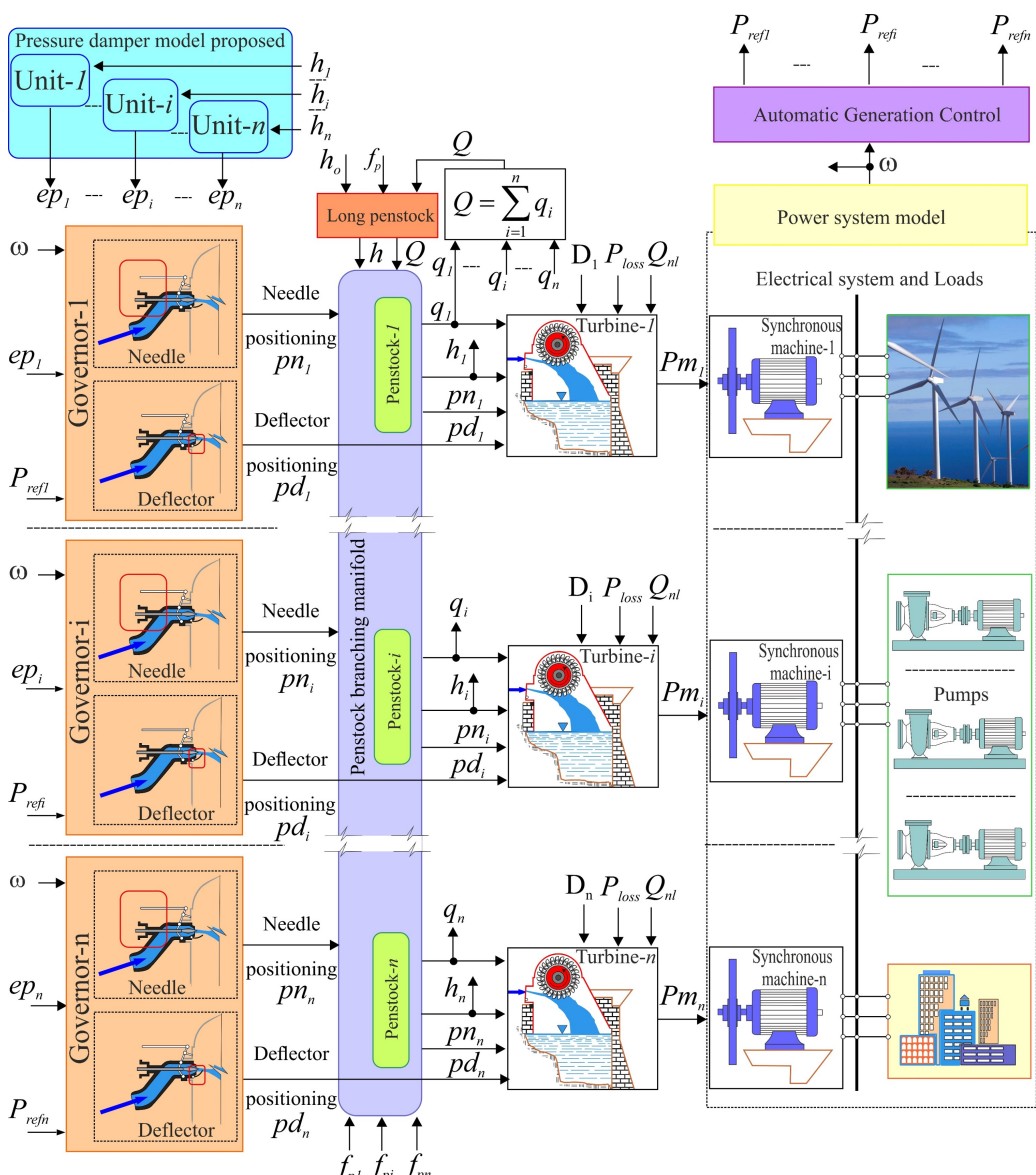

**Figure 3.** Outline of the functioning of the overall system.

Each governor-*i* emits a needle positioning signal ($pn_i$) and a deflector positioning signal ($pd_i$). The $pn_i$ signal acts regulating the water flow ($q_i$) which feeds turbine-*i*. The sum of the *n* flows $q_i$ indicates the flow *Q* of water that the long penstock has to supply to the penstock branching manifold (represented by the vertical box which contains the *n* penstocks). Input variables to the penstock branching manifold model are the aforementioned flow *Q*, the net head *h* estimated with the long penstock model and the friction head loss coefficient $f_{pi}$ in each branch pipe *i*. The penstock which pipes water from the upper reservoir to the penstock branching manifold from where it is distributed among the various turbines is assumed to be long in length, as is common for conduits used with Pelton turbines [43]. In this context, conduit elasticity and water compressibility are taken into account [44]. The pressure $h_i$ of the water at the outlet of penstock-*i* that emerges from the manifold (which is modelled considering a non-elastic water column [45]), is the input variable for the control algorithm proposed in this paper. The model of hydraulic turbine *i* is fed with the variables ($h_i$, $q_i$, $pn_i$, $pd_i$, $D_i$, $P_{loss}$ and $q_{nl}$) and provides the mechanical power $Pm_i$. In the power system model, frequency variation is dependent on the mismatch

between the power supplied by the hydroelectric units and the power provided by the non-dispatchable wind farm and the power consumed by the pumping system and the power consumed by the external loads. On this basis, estimation is made of the reference powers of each hydraulic generation unit according to their gains. The variables used in the model are defined in the different tasks that the proposed algorithm covers.

*2.3. Tasks Covered by the Proposed System Modelling Algorithm*

Figure 4 shows a block diagram with the description sequence of the set of proposed tasks to model the system. In the first task, the manifold head, *h*, is modelled. The manifold head is the sum of the travelling wave, $h_e$, which moves along the penstock, the loss in the penstock, $h_l$ (which depends on the flow, Q, and a loss factor $f_p$), and the static pressure (defined by the gross head) of the water column, $h_o$. In addition to those indicated in Figure 3, the model input data also include acceleration due to gravity, geometric (length, internal diameter, wall thickness) and mechanical (Young's modulus) characteristics of the penstock and characteristics of the water (bulk modulus, density). This task is developed in Section 2.3.1.

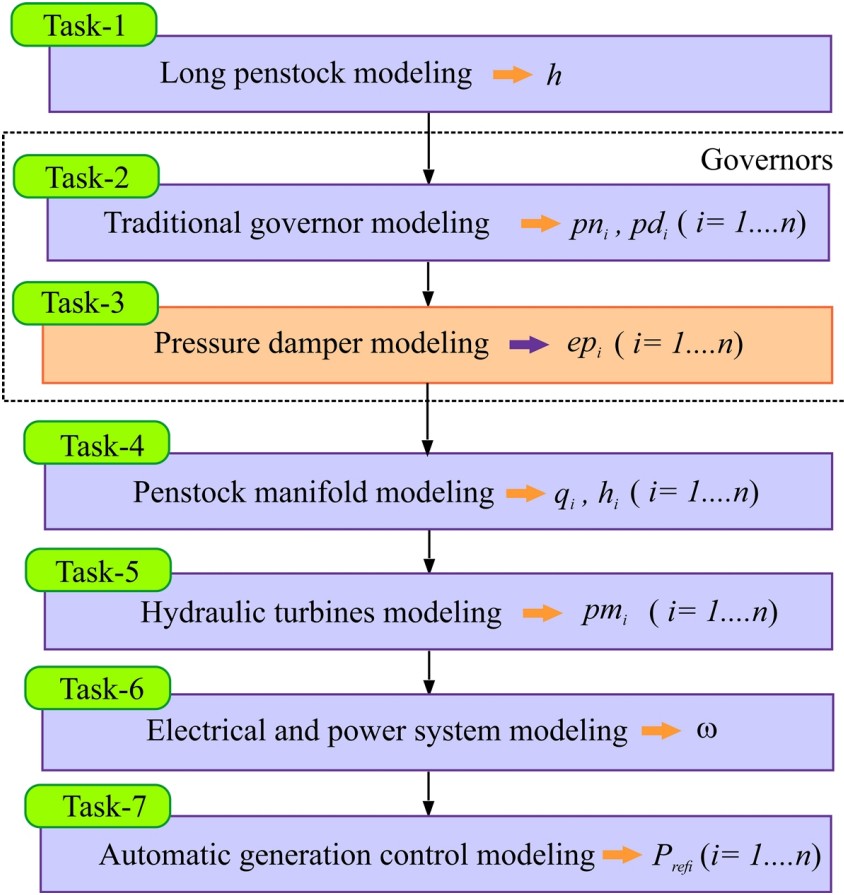

**Figure 4.** Block diagram of the proposed tasks.

The second task consists of modelling each *i* governor that has to control the mechanical power of each *i* turbine. The mechanical power of an impulse turbine is proportional to the flow of water that hits the roller and the pressure that this flow exerts on it. Two devices can be used to regulate the mechanical power of the Pelton turbine, needles and deflectors (Figure 1), which operate in accordance with the theorem of the moment [44]. Traditional impulse turbine control strategies use a PID controller circuit for the deflectors as the main turbine control device, with the needles following the bidimensional ratio between needle and deflector. However, simulation studies have shown that this approach would not

provide an acceptable frequency control performance in systems like the one analysed in the present work [46]. In this work, each $i$ governor uses two PI control loops. One PI governor loop of turbine $i$ regulates needle position $pn_i$ and the other deflector position $pd_i$ [46,47] (Figure 3).

The two PI loops are related in such a way that, except in situations when the unit needs to be protected from excess speed, the control operates in water-saving control mode. For this, the control system has to position the deflectors close to the edge of the current of water, thereby allowing all the jet emerging from the nozzle to impact the turbine buckets and generate the required torque and power output. Positioning the deflectors tangential to the water jet (following the bidimensional needle/deflector curve) allows a reduction in the time required to reposition them when required.

The deflectors intervene in situations in which there is a need to control excess speed, interrupting part or all of the water jet and diverting it from the turbine buckets and in this way eliminating, depending on the deflector position $pd_i$, part or all of the energy available in the water that emerges from the nozzle, which in turn depends on the needle position $pn_i$. The aim of this control strategy is to increase PHES operating efficiency in terms of reducing load shedding. This task is described in Section 2.3.2. Given that a long penstock is assumed, the so-called water hammer phenomenon can be significant when needle-related control tasks are undertaken, especially if high constants are used in the needle control loops to rapidly reposition them. To mitigate the impact of excess pressure, the inclusion is proposed in this work of pressure dampers (Figure 3), which are modelled in task 3, described in Section 2.3.3. The fourth task consists of modelling the manifold which receives the flow Q from the long penstock which distributes it among the $n$ penstocks that take the water to the $n$ hydraulic turbines. This model is developed in Section 2.3.4. In the fifth task, developed in Section 2.3.5, modelling of the turbine mechanical power is performed, and in the sixth task, in Section 2.3.6, of the electrical and power system. In task 7, as is widely known, estimation is made of the powers generated by each hydraulic generation unit according to their gains $R_i$. That is, the amount of load pickup on each unit is proportional to the slope of its droop characteristic [48].

### 2.3.1. Task 1: Long Penstock Modelling

The classical proposal of travelling waves [44], which considers water compressibility and conduit elasticity, was used to model the long penstock, Equation (1):

$$h_e(s) = -Q(s) \cdot Z_o \cdot \tanh(T_e \cdot s) = -Q(s) \cdot Z_o \cdot \frac{1 - e^{-2 \cdot \frac{L}{a_w} \cdot s}}{1 + e^{-2 \cdot \frac{L}{a_w} \cdot s}} \tag{1}$$

where $Z_o$ represents penstock impedance and is given by Equation (2):

$$Z_o = \frac{T_w}{T_e} \tag{2}$$

where $T_e$ is wave travel time and $T_w$ is the water time constant in the penstock, which is the time required to accelerate the water flow from zero to the flow $Q_b$ under the base head $H_b$. Both times are given by Equation (3):

$$T_e = \frac{L}{a_w} = \frac{L}{\sqrt{\frac{g}{\alpha}}} = \frac{L}{\sqrt{\frac{g}{\rho \cdot g \cdot (1/\kappa + d/f \cdot E)}}} \qquad T_w = \frac{4 \cdot Q_b \cdot L}{\pi \cdot d^2 \cdot H_b \cdot g} \tag{3}$$

The variables of Equation (3) are defined in the nomenclature table.

In the estimation of net head $h$ (Equation (4)), in addition to the contribution of the pressure waves $h_e$ and the water level $h_o$ in the upper reservoir, consideration is also given to head losses due to frictional resistance of the conduit, $h_l$, which is assumed to be

proportional to the square of the flow $Q$, with a friction head loss coefficient $f_p$, Equation (5). The friction losses in the conduit cause the oscillations to decay [36].

$$h = h_o - h_l + h_e \tag{4}$$

$$h_l = f_p \cdot Q^2 \tag{5}$$

A block diagram of the penstock simulation is shown in Figure 5 [41]. The penstock simulation was undertaken in this paper using the basic functions "Gain", "Product", "Add" and "Sum", extracted from the "Commonly Used Blocks" library of MATLAB-Simulink. The function "Transport delay" was also used, extracted from the "Continuous" library of MATLAB-Simulink, for the incorporation of the exponential term that can be seen in Figure 5.

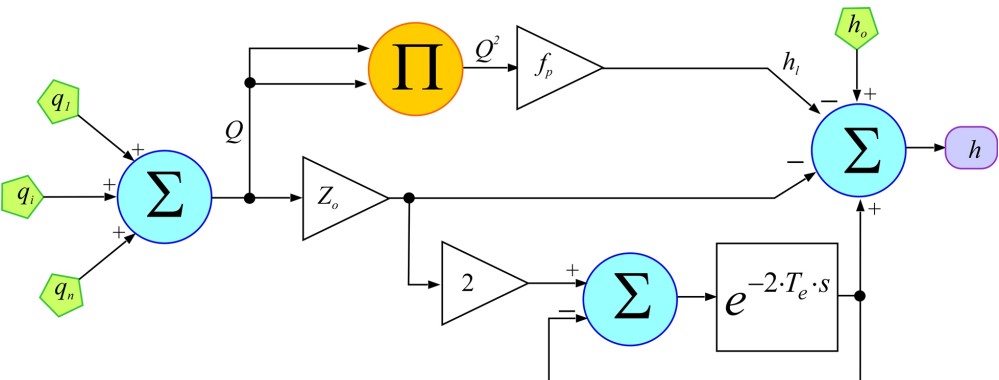

**Figure 5.** Block diagram of penstock simulation.

### 2.3.2. Task 2: Governor Modelling

Figure 3 shows the proposed PI governors for each of the $n$ turbines that make up the system and which control needle position ($pn_1, \ldots pn_i, \ldots , pn_n$) and deflector position ($pd_1, \ldots , pd_i, \ldots , pd_n$). These variables take the value 1 for the nominal position and 0 when fully closed. In the model, just one needle is considered per Pelton turbine. However, if there are multiple needles it is proposed that the servomotor of each be assumed to be identical, and that they be modelled as a single system [34].

The position $pd_i$ of deflector $i$ does not affect the flow $q_i$ which needle $i$ emits (Figure 3). However, it can affect the flow of water emitted by needle $i$ that is exploited by turbine $i$. Thus, this position $pd_i$ and the position $pn_i$ of needle $i$ influence the mechanical power $Pm_i$ of turbine $i$ (see Section 2.3.4). The position $pn_i$ of needle $i$ is used to estimate the flow $q_i$ that is introduced into penstock $i$ that takes the water to turbine $i$, and intervenes in estimation of turbine $i$ inlet pressure (see Section 2.3.3).

Figure 6 shows the action protocol of the governors. When the power generated by the WF and the PHES is greater than the power demanded by the loads, the electrical frequency of the system tends to increase. Under this circumstance, the turbine speed regulators have to act on the nozzle needles to decrease the emitted flow and in this way reduce the mechanical power $Pm_i$, (Figure 3) in order to be able to satisfy demand in an acceptable electrical frequency margin. Closure of nozzle needles in PHES with long penstocks and impulse turbines cannot result in rapid reductions in water velocity [30] because this would result in oscillations of the pressure $h_i$ in the inlet of turbine $i$ (Figure 3).

The use of pressure dampers is proposed in this work to avoid this occurring, the modelling of which is described in Section 2.3.3. In the event of excess speeds or the need for greater mitigation of the pressure waves generated during needle control, intervention of the deflectors will be activated (Figure 6). In situations in which the power generated by the WF and the PHES is less than the power demanded by the loads, the electrical frequency of the system tends to decrease. Under this circumstance, the turbine speed regulators, if they have a sufficient margin of manoeuvre, must act on the nozzle needles to

increase the emitted flow and in this way increase the mechanical power ($Pm_i$) in order to be able to satisfy demand in an acceptable electrical frequency margin. Opening the nozzle needles results in pressure reductions $h_i$ in the penstocks that take the water to the turbines. In this situation, the action of the pressure dampers has to be restricted and the deflectors should be positioned as close to the edge of the water jet as possible using a bidimensional function based on the position of the servomotor of the needle. A schematic representation of the governors is shown in Figure 7.

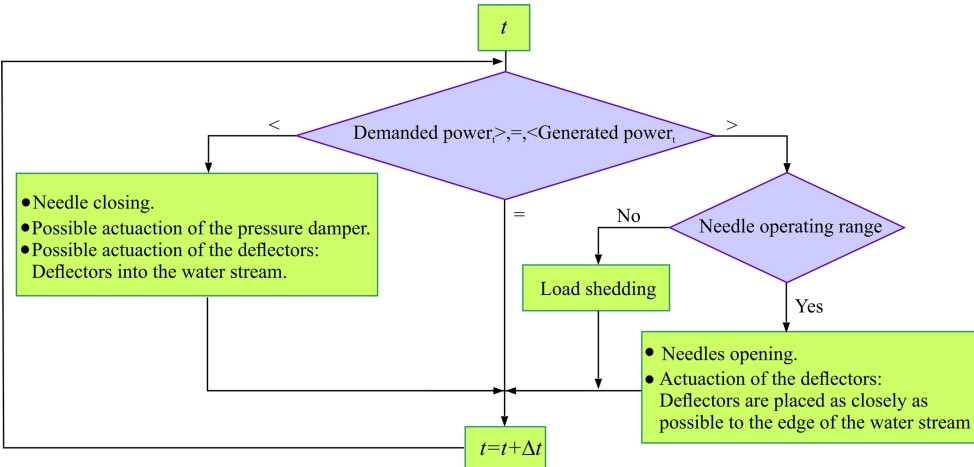

**Figure 6.** Control system action protocol in each instant $t$.

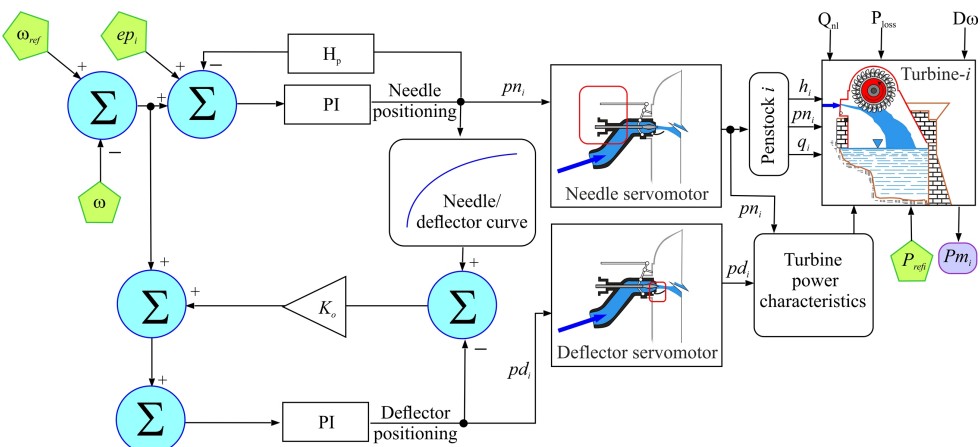

**Figure 7.** Schematic outline of governors with dual PI control loop strategy for Pelton turbines.

To optimize the dynamic operating characteristics of the turbine governor, it has separate PI gains to operate with the deflectors in and outside the water jet [46]. In this way, the servomotors of the deflectors and the needles can act simultaneously in the face of a frequency perturbation.

Equations (6) and (7) show the error $e_n(s)$ and the PI that regulates needle position, respectively:

$$e_n(s) = \omega_{ref} - \omega - pn_i \cdot H_p + e_{pi} \tag{6}$$

$$K_{PI}(s, pn_i) = K_P(pn_i) + \frac{K_I(pn_i)}{s} \tag{7}$$

Equations (8) and (9) show the error $e_d(s)$ and the PI that regulates deflector position, respectively:

$$e_d(s) = \omega_{ref} - \omega + K_0 \cdot (-pd_i + pn_i \cdot m) \tag{8}$$

$$K_{PI}(s, pd_i) = K_P(pd_i) + \frac{K_I(pd_i)}{s} \tag{9}$$

where $K_p(\cdot)$ is the proportional gain and $K_I(\cdot)$ the integral gain.

The idea is that these parameters can increase the response time of the generation groups while maintaining system stability. In this context, the aim is for the response to be damped, but as rapidly as possible. In some systems, this may be insufficient and slow response of hydro turbines in the face of wind ramps or renewable generation ramps can generate frequency variations on a regular basis.

To estimate the parameters of the PI governor models, a criterion is proposed based on root locus analysis. In the case of constructed systems in operation, it is proposed, as also done by other authors [38], to tune the parameters used to optimize the control strategy. The regulator incorporates an inertia constant $H_p$ in steady state. The simulation of governors with dual PI control loop strategy for Pelton turbines was undertaken using the basic functions "Gain", "Product", "Add", "Sum", "1-D Lookup Table" and "Transfer Fcn", extracted from different commonly used libraries of MATLAB-Simulink and the function "Hydraulic Turbine and Governor" of the "Simscape/Electrical/Specialized Power Systems/Fundamental Blocks/Machines" library of MATLAB-Simulink [49].

### 2.3.3. Task 3: Proposed Pressure Damper Modelling

With the aim of damping pressure wave oscillations as a result of the rapid action of the speed regulator, it is proposed to use the nozzle needle of each turbine as a relief valve. More specifically, when the pressure rises the idea is for the nozzle needle to open further and in this way obtain a pressure reduction, and when pressure falls for the needle to tend to close and in this way obtain a pressure increase.

Given that the nozzle is governed by the speed regulator and that this regulation should not be compromised, it is proposed that speed regulation and the proposed damping effect be coordinated. For this, a pressure error signal $ep_i$ is added to the PI governor input of turbine $i$, which depends on the turbine input pressure $h_i$, and a pressure that is taken as reference $p_{ref}$, which represents the pressure of the conduit working under steady state if it were not affected by the oscillation due to the transient state (Figure 3). Then, if $h_i < p_{ref}$ (pressure decrease, $ep_i < 0$) the nozzle needle tends to open and if $h_i > p_{ref}$ (pressure increase, $ep_i < 0$) it will tend to close, with the aim of damping the pressure oscillation through these actions.

The routine to generate the $ep_i$ consists of two blocks, named detector and actuator (Figure 8).

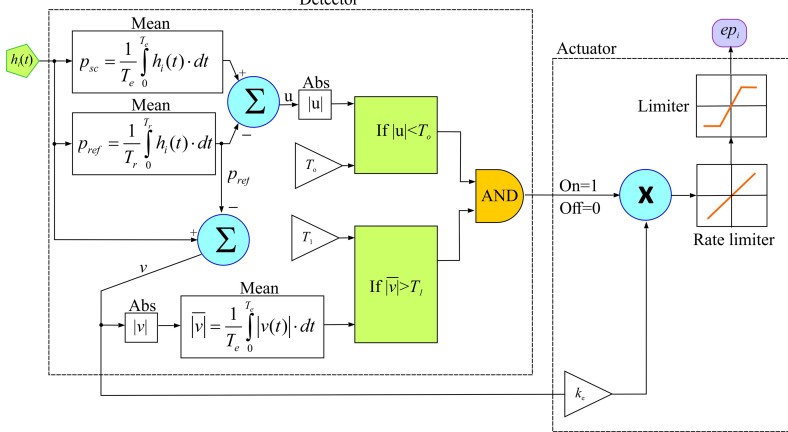

**Figure 8.** Block diagram of the detection and actuation functions of the damper.

While various detection methods are possible, in this study it has been designed, as can be seen in Figure 8, with the aim of facilitating its adaptation to the mathematical functions and programming capacities of commercial speed regulators.

The mission of the detector block is: (a) to determine whether an oscillation of $h_i$ is being produced with respect to $p_{ref}$, and (b) to analyse whether the amplitude of that oscillation is high. Only if both conditions (a) and (b) are met will the detector block set in motion the actuator block.

In the analysis process of condition (a), the detector block compares $p_{ref}$ with a pressure $p_s$. $p_s$ is the mean of $h_i$ generated during the wave travel $T_e$. $p_{ref}$ is defined as the mean of the $h_i$ generated during a time $T_r > T_e$. It is considered that condition (a) is met if $\left| p_s - p_{ref} \right| < T_o$, where $T_o$ is a predetermined bound value (Figure 8).

The detector block considers that condition (b) is satisfied if the mean, during a time $T_e$, of the variable $\left| h_i - p_{ref} \right|$, is higher than a predetermined parameter $T_1$ (Figure 8).

The actuator block, when activated by the detector block (signal on), generates the error signal $ep_i$, which will have a null value when the oscillation is acceptable and a non-null value when the detector block finds that an oscillation is present that needs to be damped.

The pressure error is calculated as the difference between the instantaneous pressure and the pressure $p_{ref}$. This value is weighted with a constant $k_e$, used to adjust the influence of the pressure error in the speed regulation loop. The damper incorporates a rate limiter which mitigates sudden changes in $ep_i$, especially during its activation (on) and deactivation (off). Finally, a limiter is also included which restricts actuation when the pressure errors are negative. This is taken into consideration because if the nozzle needle closes when there is a pressure fall there will be a decrease in flow and hence a reduction in the mechanical power of the turbine, which can provoke under-frequencies. In this context, it is considered preferable to act especially with positive pressure errors, opening the nozzle. This will cause an increase in the flow to the turbine which could generate over-frequencies. However, these can be more easily mitigated through intervention of the turbine deflector. The simulation of the block diagram of the detection and actuation functions of the damper was programmed with the basic functions "Gain", "Product", "Add", "Mean", "Sum", "Relational Operator", "Abs", "Logical Operator" and "Saturation", extracted from different libraries of MATLAB-Simulink.

### 2.3.4. Task 4: Penstock Branching Manifold Modelling

In this case, the modelling is based on the assumption that the branch pipes (or penstocks) which transport the water to the turbines from the manifold are short in length. In this context, water compressibility is assumed to be negligible for the calculations.

Likewise, it is considered that the flow Q which travels through the long penstock is the same as the flow that enters the manifold [45]. The flow $q_i$ which enters penstock $i$ (*i*th penstock) of the manifold is estimated through Equation (10), [44]:

$$q_i = Q \cdot An_i \cdot \sqrt{h} \; ; \; Q = \sum_{i=1}^{n} q_i \tag{10}$$

where $An_i$ is a function of the position $pn_i$. On the basis of an analysis of experimental data obtained by different authors [42,43], this has been identified as a quadratic function of position $pn_i$. $h$ is the head at the manifold inlet and is an output variable of the long penstock model (Figure 3). Equation (11) records the flow dynamic in each branch pipe $i$, including the hydraulic interaction (coupling effect) with the other turbines. The start times of the water in the branch pipes are given by $Tw_l, \ldots, Tw_i, \ldots, Tw_n$, and the start

time of the water in the manifold by $Tw$, Equation (3). The model is deduced from the basic moment equations for each branch pipe [44–48,50,51].

$$
\begin{bmatrix} \dot{q}_1 \\ \cdots \\ \dot{q}_i \\ \cdots \\ \dot{q}_n \end{bmatrix} = \begin{bmatrix} Tw + Tw_1 & \cdots & Tw & \cdots & Tw \\ \cdots & \cdots & \cdots & \cdots & Tw \\ Tw & \cdots & Tw + Tw_i & \cdots & Tw \\ \cdots & \cdots & \cdots & \cdots & Tw \\ Tw & Tw & Tw & Tw & Tw + Tw_n \end{bmatrix}^{-1} \cdot \begin{bmatrix} h - h_1 - h_{l1} \\ \cdots \\ h - h_i - h_{li} \\ \cdots \\ h - h_n - h_{ln} \end{bmatrix}
$$

$$
\begin{bmatrix} \dot{q}_1 \\ \cdots \\ \dot{q}_i \\ \cdots \\ \dot{q}_n \end{bmatrix} = \begin{bmatrix} T_{11} & \cdots & T_{1j} & \cdots & T_{1n} \\ \cdots & \cdots & \cdots & \cdots & \cdots \\ T_{i1} & \cdots & T_{ij} & \cdots & T_{in} \\ \cdots & \cdots & \cdots & \cdots & \cdots \\ T_{n1} & \cdots & T_{nj} & \cdots & T_{nn} \end{bmatrix}^{-1} \cdot \begin{bmatrix} \Delta h_1 \\ \cdots \\ \Delta h_i \\ \cdots \\ \Delta h_n \end{bmatrix} \tag{11}
$$

In Equation (11), $\dot{q}_i = dq_i/dt$, $h_{li}$ is head loss due to friction in branch pipe $i$, which is assumed to be proportional to the square of the flow $q_i$, Equation (12), with a friction head loss coefficient in branch pipe $i$ of $f_{pi}$. $h_i$ is the inlet pressure of turbine $i$ [43], Equation (12):

$$
h_{li} = f_{pi} \cdot q_i^2 \; ; \; h_i = \left( \frac{q_i}{An_i} \right)^2 \tag{12}
$$

A block diagram of the penstock branching manifold modelling is shown in Figure 9. The simulation of this block diagram was programmed with the basic functions "Gain", "Product", "Sum" and "Integrator", extracted from different libraries of MATLAB-Simulink.

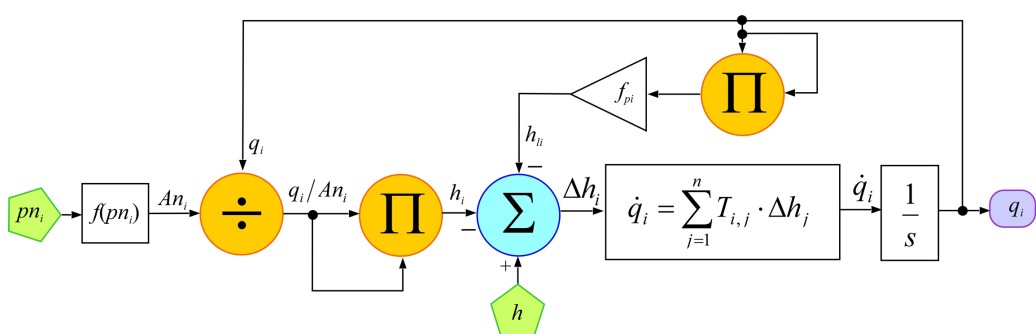

**Figure 9.** Block diagram of penstock branching manifold.

2.3.5. Task 5: Hydraulic Turbine Modelling

The hydraulic power of the needle jets of turbine $i$ is given by Equation (13) [45,47]:

$$
P_{wi} = (q_i - q_{nl}) \cdot h_i \cdot A_t \tag{13}
$$

In Equation (13), $q_{nl}$ represents the flow without head or the minimum flow necessary for the turbine to provide useful power. $A_t$ is the turbine gain and is given by Equation (14) [52]:

$$
A_t = \frac{TRP}{GRP \cdot h_r \cdot (q_r - q_{nl})} \tag{14}
$$

where $h_r$ is the head per unit in the turbine at nominal flow, $q_r$ is the flow per unit at nominal head, $TRP$ is the turbine MW rating and $GRP$ the generator base MVA [52].

The input power $P_{ti}$ to turbine $i$ is estimated through Equation (15) [48]:

$$
P_{ti} = P_{wi} \cdot f(pn_i, pd_i) \tag{15}
$$

where $f(p_{ni}, p_{di})$ is a non-linear function of the variables $p_{ni}$ and $p_{di}$, (Figure 7).

The mechanical power of turbine *i* is determined through Equation (16) [45,52]:

$$P_{mi} = P_{ti} - P_{loss} - P_{damping} = P_{ti} - P_{loss} - D_\omega \cdot An_i \cdot \Delta\omega \tag{16}$$

where the term $P_{damping}$ takes into account the damping effect due to friction and is proportional to rotor speed deviation $D_w$ and needle opening $An_i$.

A block diagram of the modelling of the mechanical power of turbine *i* [34,53] is shown in Figure 10. The simulation of this block diagram was programmed with the basic functions "Gain", "Product", "Sum" and "1-D Lookup Table", extracted from different libraries of MATLAB-Simulink.

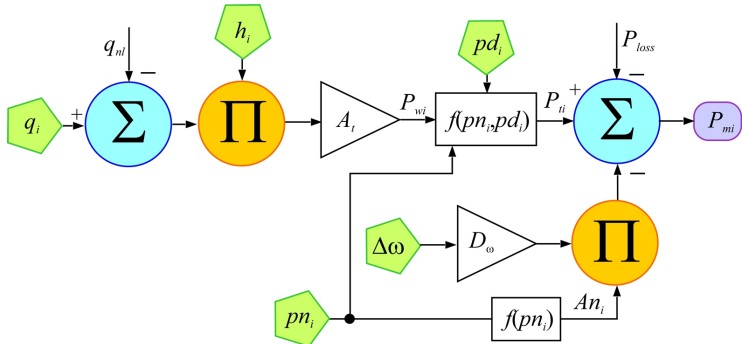

**Figure 10.** Block diagram of estimation of the mechanical power of a turbine.

### 2.3.6. Task 6: Electrical and Power System Modelling

For the analysis of load-frequency controls of the isolated system, it is assumed that this is of small size and the use is proposed of a single-bus model [52]. This simplification has been considered by other authors who have focused on a small island [38–41] and a relatively large island [54]. Here, Equation (17) is used in which frequency variation is dependent on the mismatch between the power $P_{HT}$ supplied by the hydroelectric units and the power $P_{WF}$ provided by the non-dispatchable WF and the power $P_p$ consumed by the pumping system and the power $P_L$ consumed by the external loads (Figure 3).

$$f\frac{df}{dt} = \frac{1}{2H}\left(\sum_{i=1}^{n} P_{mi} + P_{WF} - P_p - P_L - D_{net} \cdot \Delta f\right) \tag{17}$$

In the model, neither oscillations between hydroelectric units nor the efficiency of the transmission system are taken into account. That is, a coherent response of all units to changes in the system load is assumed. Therefore, an equivalent generator is considered with an inertia constant $M_{eq} = 2H$ equal to the sum of the inertia constants of all the hydroelectric units [44]. It is considered that this equivalent generator is driven by the combined mechanical powers of the *n* individual hydro turbines. The sensitivities of the loads to frequency variations are grouped into a single damping constant $D_{net}$ (Figure 11). The simulation of the system equivalent for load-frequency control block diagram was performed with the basic functions "Gain", "Sum" and "Transfer Fcn", extracted from different libraries of MATLAB-Simulink.

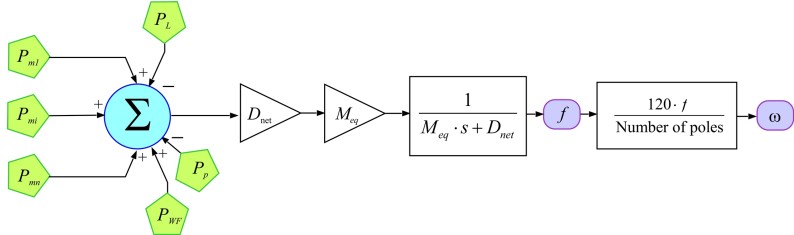

**Figure 11.** System equivalent for load-frequency control.

## 3. Case Study

The island of El Hierro forms part of the Canary Archipelago, one of the seventeen autonomous communities of Spain, designated an outermost region of the European Union. It is situated in the Atlantic Ocean, close to the northwest coast of Africa, between latitudes 27°37′ and 29°25′ north (subtropical location) and longitudes 13°20′ and 18°10′ west of Greenwich (Figure 12). El Hierro was declared a Biosphere Reserve by Unesco in 2000. It has a surface area of 268.71 km² and a population (in 2020) of 11,147 inhabitants. On 22 November 1997, the local island government approved a Sustainability Plan with the aim of making the island self-sufficient in energy terms. In 2004, the Gorona del Viento, S.A. company was set up to develop the 'El Hierro wind–hydro power plant' project. Construction of the plant commenced in August 2009 and concluded in September 2013. The opening ceremony took place in June 2014. The island's generation system comprises 10 thermal groups (diesel engines) with unit capacities ranging between 780 and 2000 MW to make a total installed capacity of 14.9 MW, a WF with rated power of 11.5 MW (comprising 5 × 2.3 MW Enercon E-70 wind turbines), PV panels with an installed capacity of 0.03 MW and a PHES system with 4 × 2830 kW Pelton turbines (maximum flow rate of 2.0 m³/s and gross head of 655 m) and total power of 11.32 MW. That is, the technological structure of the island's generation system integrates fossil fuel-based energy through the diesel engines (39.5%) and renewable energies (60.5%). The PHES comprises a pumping system with two centrifugal pump units, each with 1500 kW variable frequency drive, and 6 × 500 kW centrifugal pump units, making a total power of 6 MW (Figure 13). The upper reservoir has a maximum capacity of 380,000 m³ and is connected to the Pelton turbines via a steel penstock 1 m in diameter and 2350 m in length. The lower reservoir has a maximum storage capacity of 149,000 m³ and is connected to the pumping system via a 3015 m long steel pipe 0.8 m in diameter.

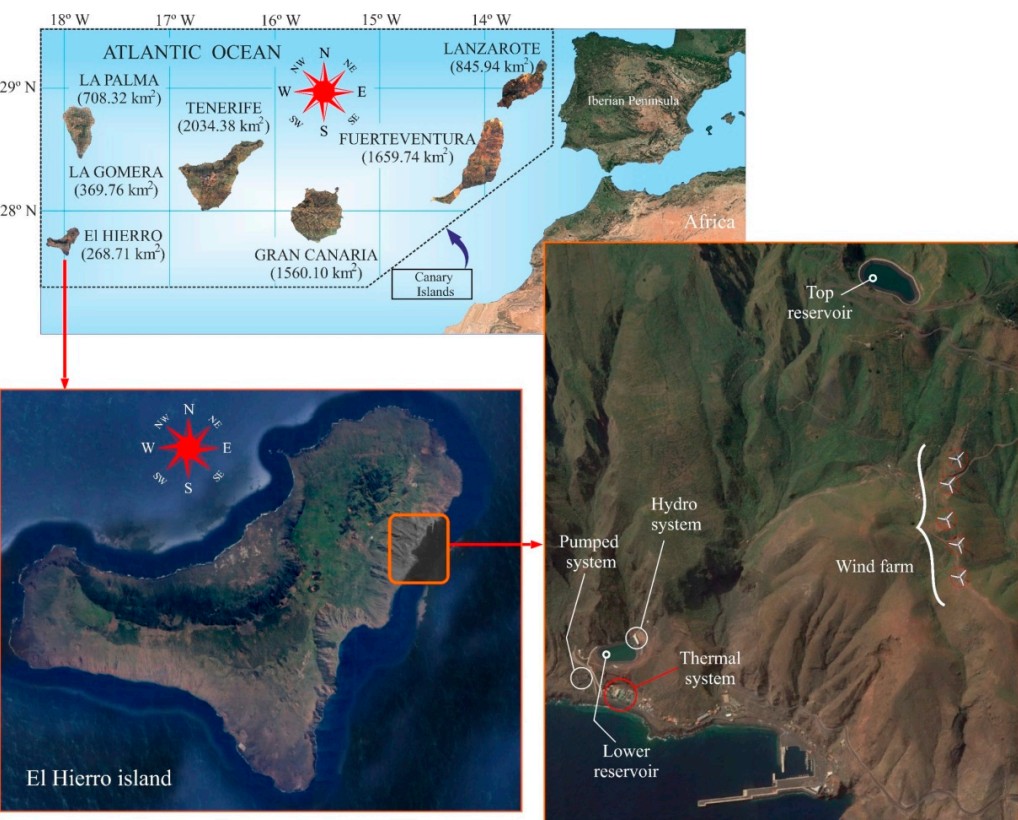

**Figure 12.** Location of the Corona del Viento wind–hydro power plant. Source: adapted from [22,23], with permission from Elsevier, 2023.

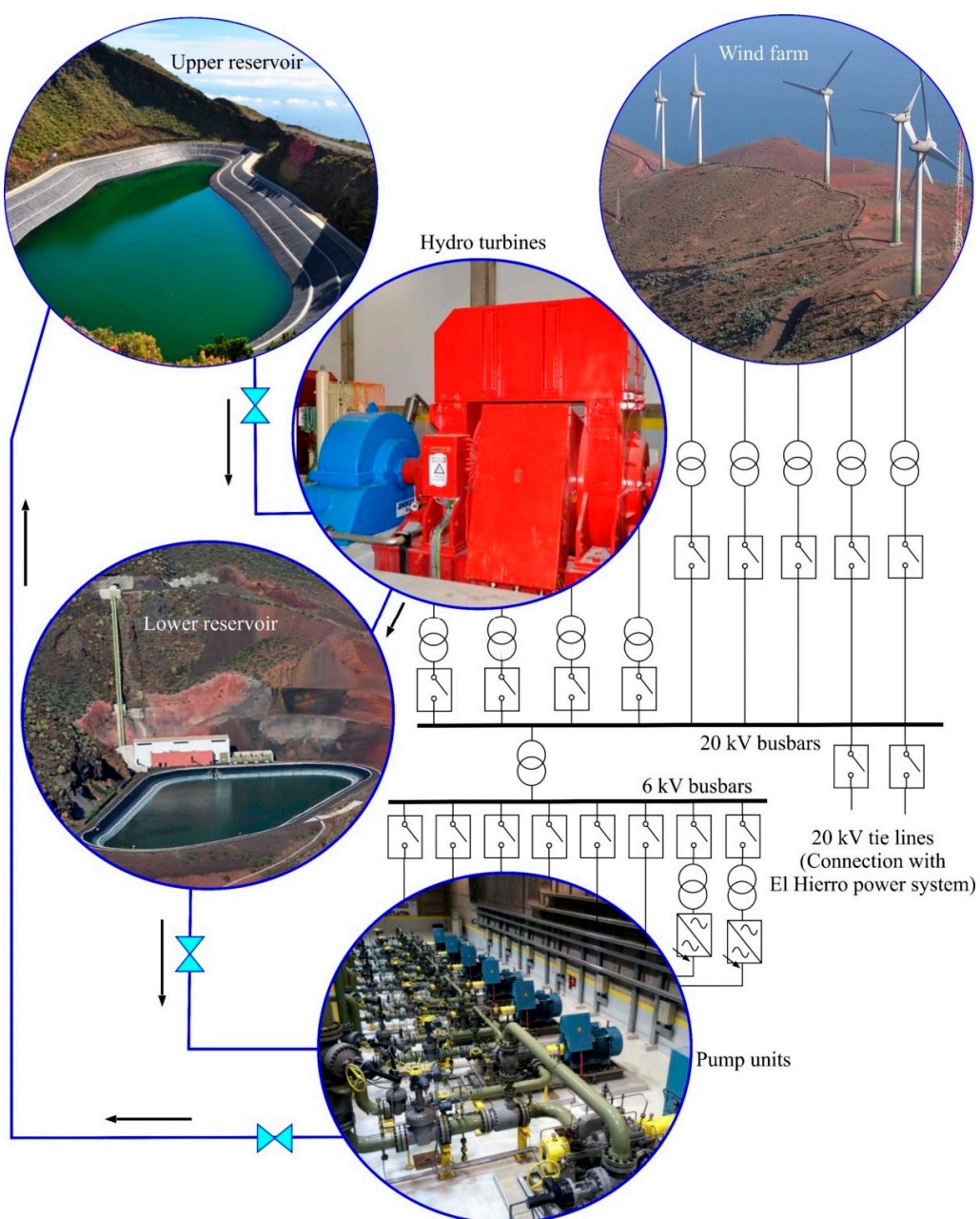

**Figure 13.** Electrical–hydraulic layout of the Corona del Viento wind–hydro power plant.

The PHES has an under-frequency pump shedding system. If the frequency reaches values below 49.3 Hz, the pump groups are disconnected which helps to stabilize the frequency of the system at acceptable values. During initialization of the PHES system, the speed regulators of the turbines were adjusted to optimize their response time. After this adjustment, the system was stable and turbine regulation was possible to a degree, but the response time was insufficient.

Until the end of 2017, an average 700 pump group shedding events were taking place each year, which gives an idea of the frequency deviations that were being generated in the system.

In November 2017, the damper described in Task 3 (Section 2.3.3) was implemented in the PHES speed regulators.

## 4. Results Analysis

In this section, a justification is provided of the periods of disconnection in the case study of the diesel groups which, as indicated by Wang et al. [42], must be analysed in order to guarantee the safe and stable operation of the electrical system. Following this, the results of the simulation of PHES operation with and without the proposed pressure damper are presented. Finally, a comparison with a real event is carried out, after implementation of the damper system in the governors of the turbines.

### 4.1. Justification of the Disconnection Periods of the Diesel Groups

As indicated in Section 1.2, the aim is to validate the proposed model in situations in which the diesel groups are disconnected, the energy from the WF and PHES is sufficient to satisfy demand, and frequency control is provided exclusively by the PHES. It was found from an analysis of the 10 min data published by REE (Red Eléctrica de España [55]), Spain's transmission system operator (TSO), that this situation occurred up to 156 times on El Hierro in the 2017–2021 period (Figure 14). In Figure 14, as well as in Figures 15–18, a legend is included to facilitate their interpretation, indicating what the different lines mean on a box and whisker diagram. Each circle represents an outlier, which is to say it represents an unusually large or an unusually small value compared to the others. The outliers indicated in the aforementioned figures indicate a peculiarity of the process under study. Further information about boxplots and their interpretation can be found in [56].

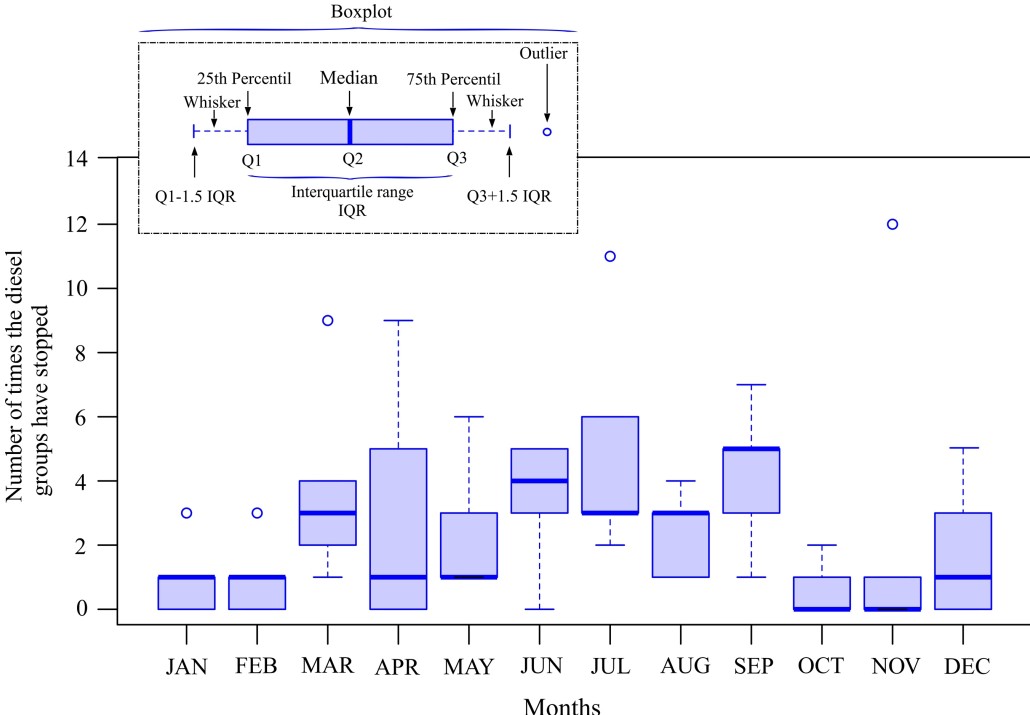

**Figure 14.** Monthly boxplot of the number of times the diesel groups were shut down in the 2017–2021 study period.

As can be seen in Figure 15, considering monthly periods, the diesel groups were inactive for up to 39,000 min in July 2018. That is, in that month and for a period equivalent to just over 27 days, frequency control was provided solely and exclusively by the PHES, and the energy from the WF and PHES covered the demand. Overall, in the 2017–2021 period, the diesel groups were inactive for up to 477,560 min, the equivalent of over 331 days. The median values of the monthly time in which the diesel groups were inactive were highest in the months of July and August (Figure 15).

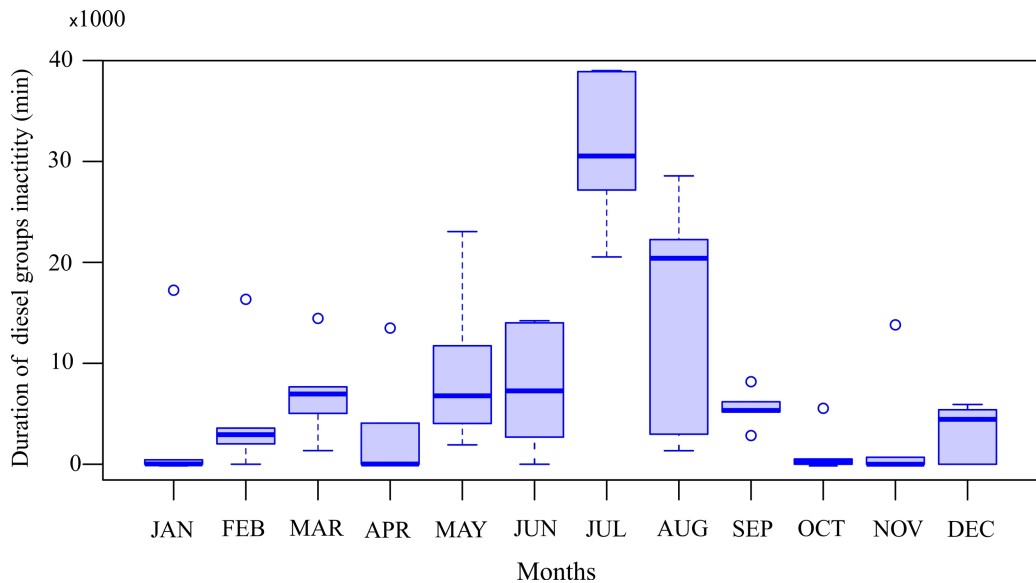

**Figure 15.** Monthly boxplot of duration (in min) of diesel group inactivity in the 2017–2021 study period.

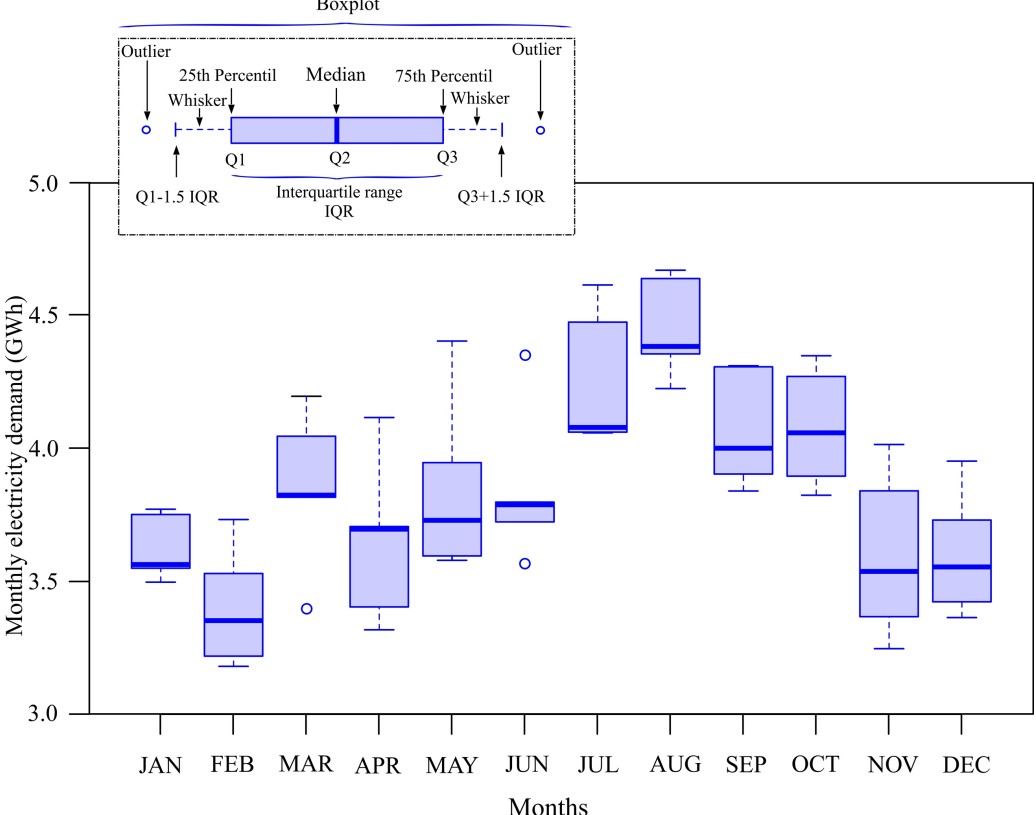

**Figure 16.** Monthly boxplot of electricity demand in the 2017–2021 study period.

One reason for this inactivity is the influence of the NE trade winds which cross the Canary Archipelago almost all year round [57]. These winds vary in intensity depending on the displacement of the Azores anticyclone over the course of the year. During most of spring and even part of autumn, the frequency of the trade winds regime is very high, but in summer it can reach up to 90–95%. The intensity of the trade winds means that although the electricity demand in these months is very high (Figure 16), so too is renewable penetration (Figure 17).

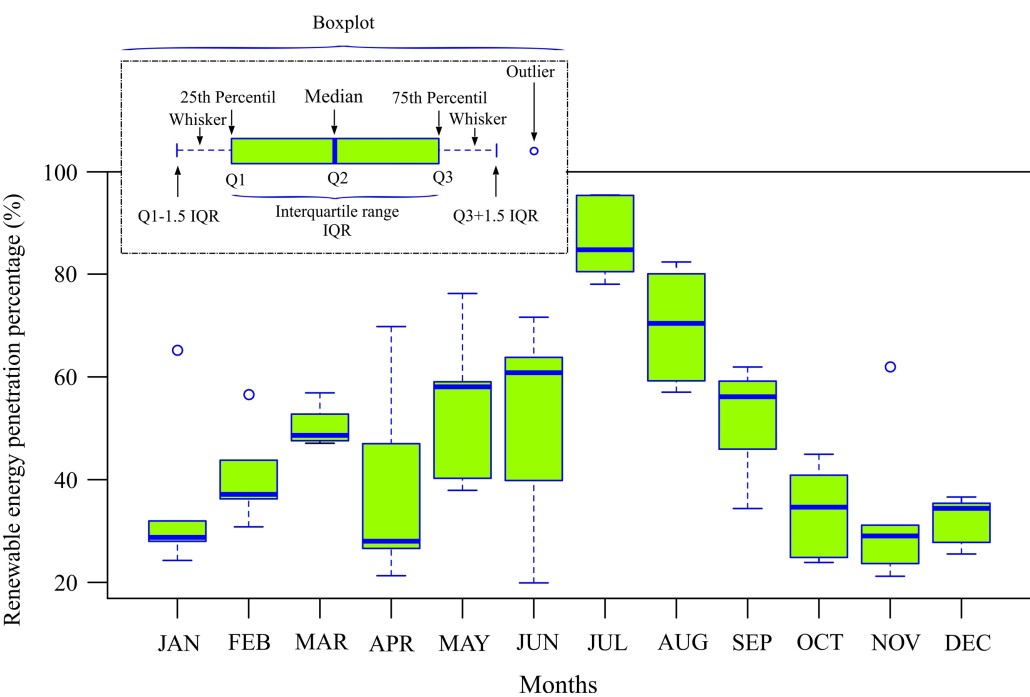

**Figure 17.** Monthly boxplot of renewable energy penetration percentage in 2017–2021.

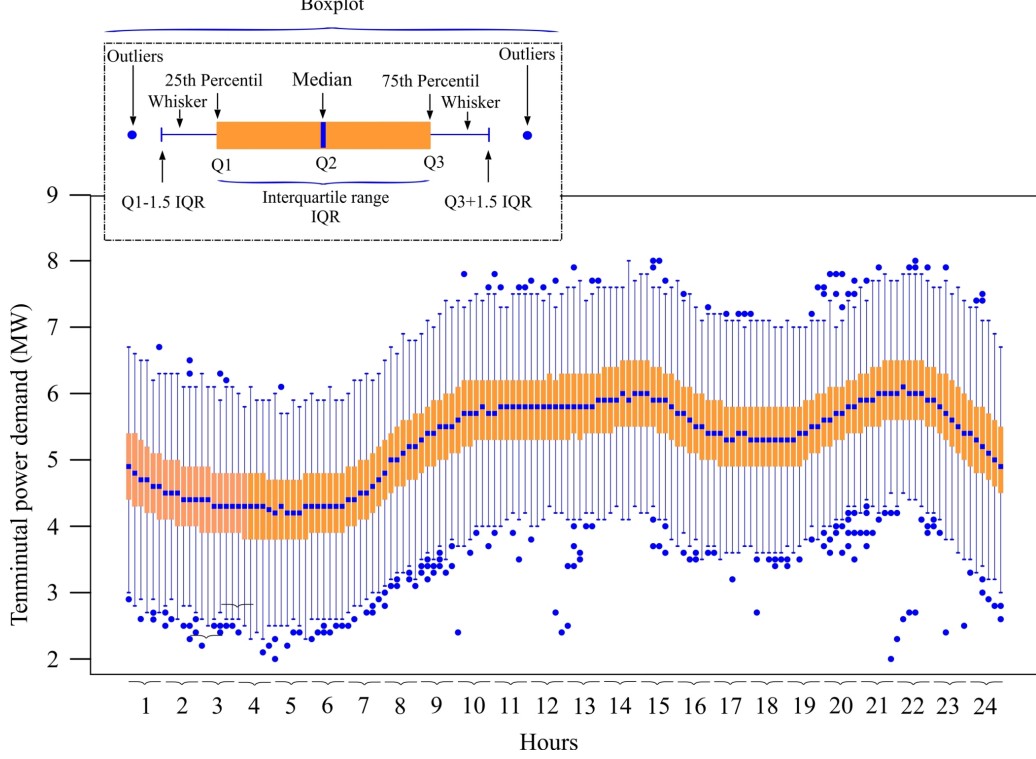

**Figure 18.** Boxplot of evolution of 10 min electrical energy demand in 2017–2021.

In the fight against climate change, the European Union has drawn up the Energy Roadmap 2050 with a view to substantially reducing the emission of pollutants into the atmosphere [1]. The energy generation sector will play a vital role in facilitating a profitable transition to a low carbon emission economy. Among other measures, this will entail replacing fossil fuels with non-polluting renewable energy sources.

In the 2017–2021 period that was analysed and according to the data used [58], renewable penetration was distributed monthly as shown in Figure 17. As can be ob-

served, July had the highest renewable penetration, reaching values of 96% and 97% in 2018 and 2019, respectively.

Generally, when covering the daily electrical energy demand with peaks of up to 8 MW (Figure 18), the electrical system responds by following the strategies established by the TSO, injecting energy generated by the different sources (diesel, wind, hydraulic). An example of this daily coverage and the different sources used is shown in Figure 19. However, the method and results considered in the present study are focused, as previously mentioned, on those periods in which the electrical energy demand is covered exclusively by wind and/or hydraulic energy. An example of such a period, covering an entire day, is provided in Figure 20.

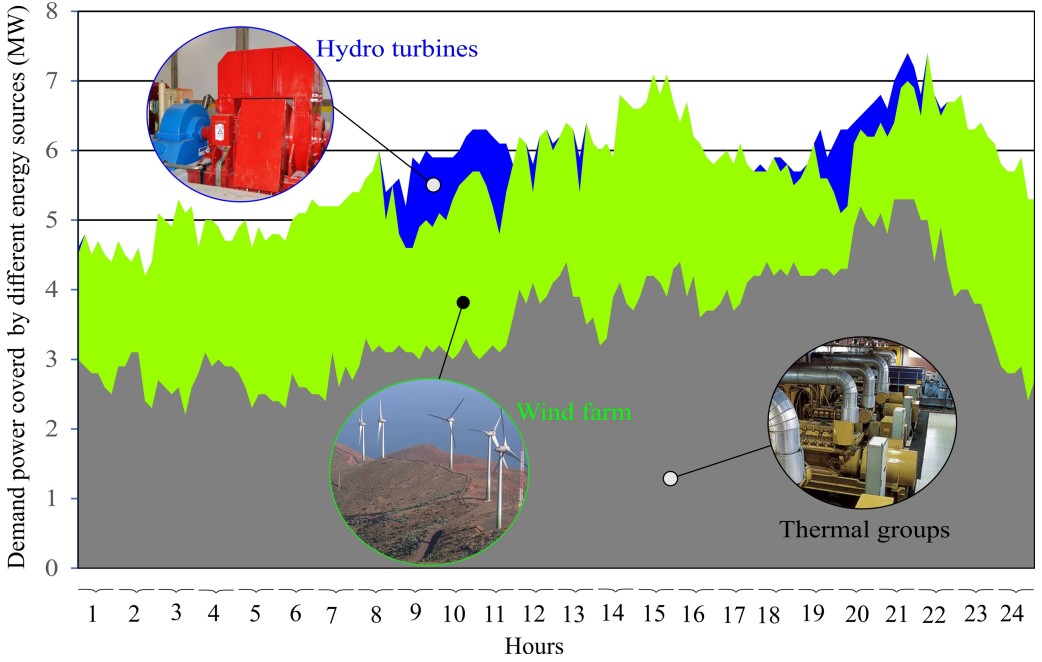

**Figure 19.** Ten-minute demand covered by different electrical energy sources (1 March 2021).

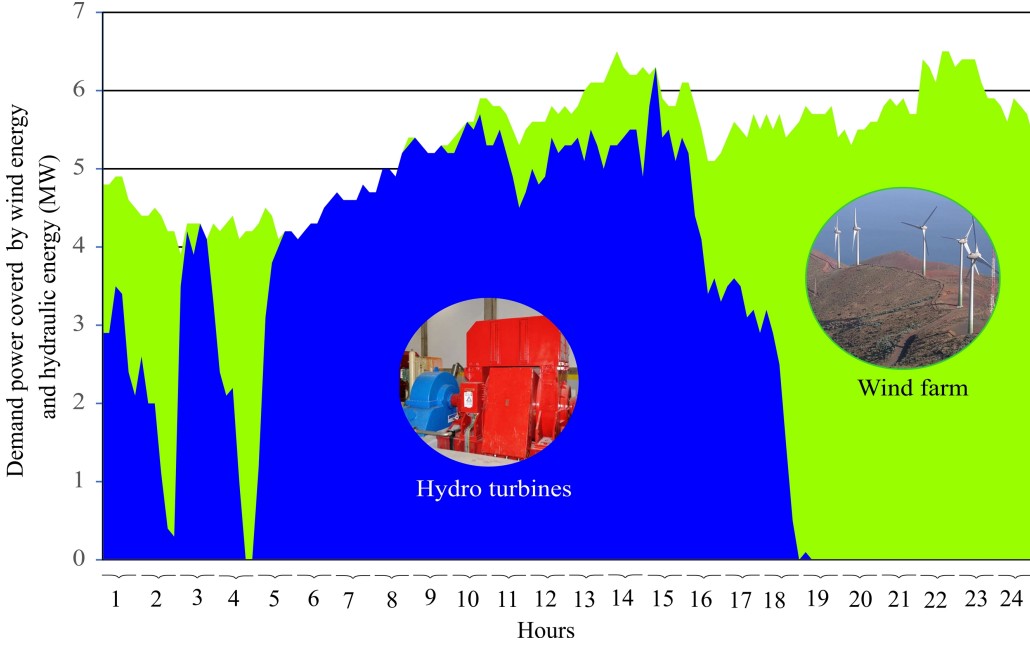

**Figure 20.** Ten-minute demand covered solely by wind and/or hydraulic energy (30 July 2018).

### 4.2. Results of the Simulation

The simulation of the evolution of the mechanical power and frequency of the system, with and without the proposed pressure damper, in the face of a loss of generation is shown in Figure 21a. The performance of the opening of the nozzle needle and pressure is shown in Figure 21. In both figures, the simulation is of the response of three turbines to a loss of generation in a 2 MW ramp that takes place in t = 20 s.

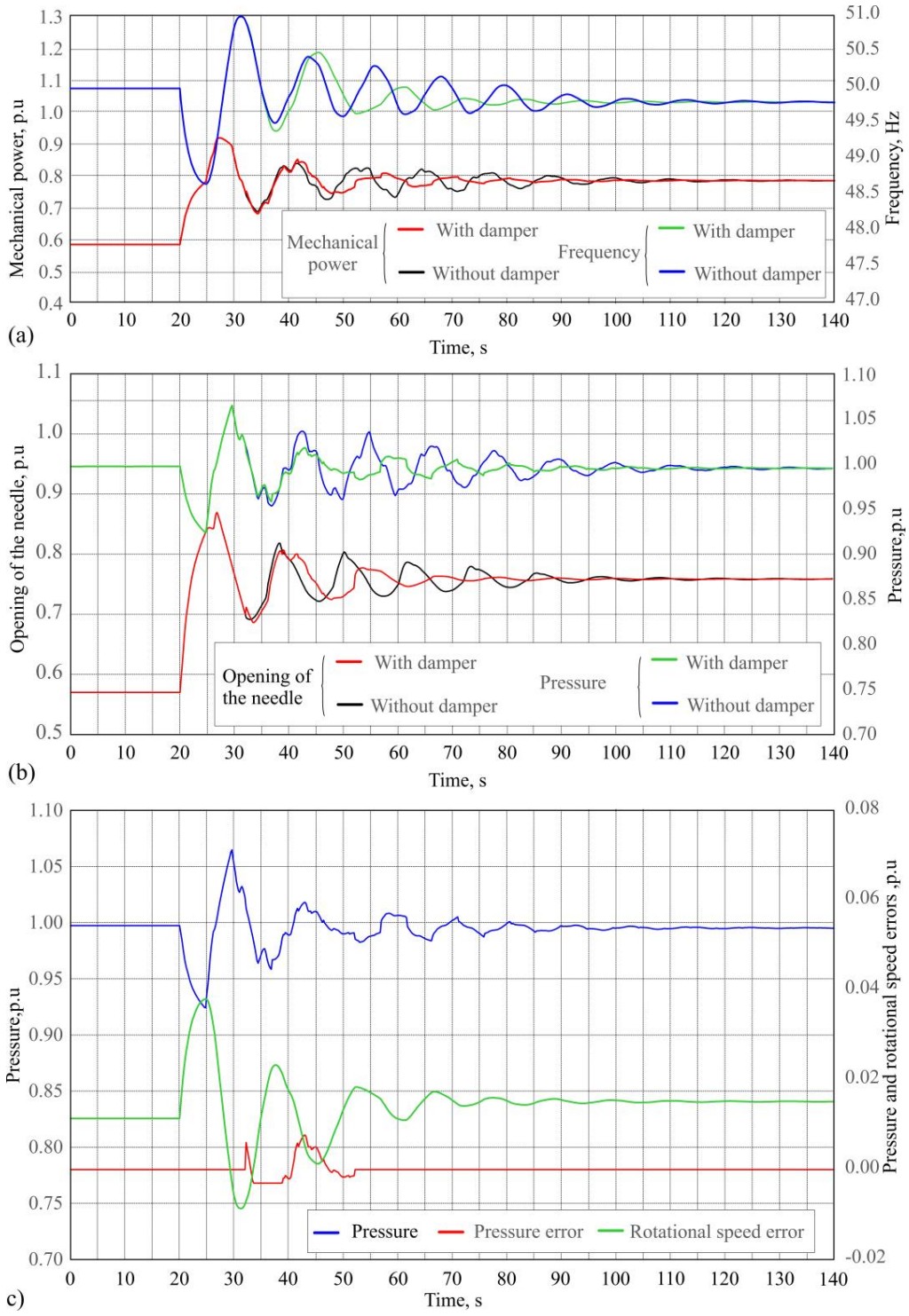

**Figure 21.** Evolution in the face of generation loss with vs. without damper of: (**a**) Mechanical power and frequency, (**b**) Needle opening and pressure, (**c**) Pressure, pressure error and rotational speed error.

From the instant at which the speed regulators detect the fall in frequency generated by the loss of generation, these act by opening the nozzle needle of the group in order to allow a greater flow (Figure 21a). As a result of this action, the system frequency is only reduced to the minimum value reached of 48.6 Hz. It can be observed in Figure 21a,b that evolution of the variables is identical up to t = 32.6 s, both when the system has and does not have the proposed pressure damper. From that instant onwards, in order to regulate the power, the control begins to close the nozzle, which generates an overpressure in the penstock. It is at this point that the system with the proposed pressure damper diverges from the system without it. The former begins to generate an error $e_p$ which affects the evolution of the opening of the nozzle needle (Figure 21b) and consequently the evolution of pressure (Figure 21b), power and frequency (Figure 21a). It can be seen in these figures that the oscillation of these variables is more rapidly damped from the instant the damper commences its intervention. As shown in Table 1, when t > 80 s, the oscillation is practically nullified in the system with damper, but continues in the system without. That is, damper implementation achieves a more stable response than that obtained by the system without damper.

**Table 1.** Comparative results of operating parameters of the system with and without pressure damper.

| Damper Available | Minimum Frequency (Hz) | Over-Frequency (Hz) | Pressure Oscillation in t = 80 s (pu) | Frequency Oscillation in t = 80 s (Hz) |
|---|---|---|---|---|
| Yes | 48.6 | 51 | 0.01 | <0.1 |
| No | 48.6 | 51 | 0.06 | 0.4 |

For the simulation with damper, the pressure in the conduit (h), the speed error ($e_v$) and the pressure error ($e_p$) are represented in Figure 21c. When the damper is activated, the pressure error ($e_p$) becomes other than zero. This takes place at t = 32.6 s, after the speed regulator has corrected the loss of generation. The action of the damper extends to t = 52.6 s.

Therefore, the damper is not activated immediately, allowing the regulator to correct the frequency. This action of regulation causes a pressure oscillation which in turn causes the damper to be activated which acts reducing the oscillation.

While the damper is in action, the pressure error ($e_p$) tends to counteract the pressure increase during the transient period when the wave oscillation takes place. When the pressure is above the initial pressure ($p_{ref}$), it tends to open the nozzle and when below it tends to close it.

The pressure error and the speed error are in phase opposition and the speed errors are, in general, larger than the pressure errors. For this reason, the tendency continues to be to regulate speed, but while the pressure error is being generated the evolution of the nozzle is modified resulting in oscillation damping. The adjustment constants of the PI where the damper intervenes allow to mitigate or increase the effect of the pressure error in such a way that it reduces the oscillation without nullifying speed regulation. It can also be observed how the pressure error is saturated for negative values. This is due to the limiter, incorporated to restrict damping at pressures below $p_{ref}$, closing the nozzle, given that this could decrease the mechanical power of the group and produce under-frequencies.

Thanks to the damper, it has been possible to increase the parameters of the PI governor in such a way that the capacity of the system to act in the face of generation trips and wind ramps is improved.

However, the increase of these PI parameters and, on occasion, the action of the damper itself, which tends to open during over-pressures, can provoke under-frequencies which are avoided through the action of the deflector. Thus, adjustment of the deflector has also been improved in such a way that it contributes assistance during the excess speeds that are produced. The deflector and its control were modelled in MATLAB-Simulink following the procedure described in Section 2.3.2.

Shown in Figure 22 are the simulations of the system operating with the damper in two different situations: (i) when its use is combined with the effect of the deflector, and

(ii) when it operates without the help of the deflector. To obtain these results, the same loss of generation was simulated as in the previous case (2 MW). During the analysis, the three hydro turbines which respond to the generation loss remained coupled. As can be seen, the over-frequency which the system reaches with the deflector (50.4 Hz) is lower than that reached without (51 Hz). Note also how the action of the deflector reduces oscillations in the response of the system in the face of the same loss of generation. Thus, it can be deduced that the effect of the deflector helps to reduce not only the maximum values of frequency but also its oscillation.

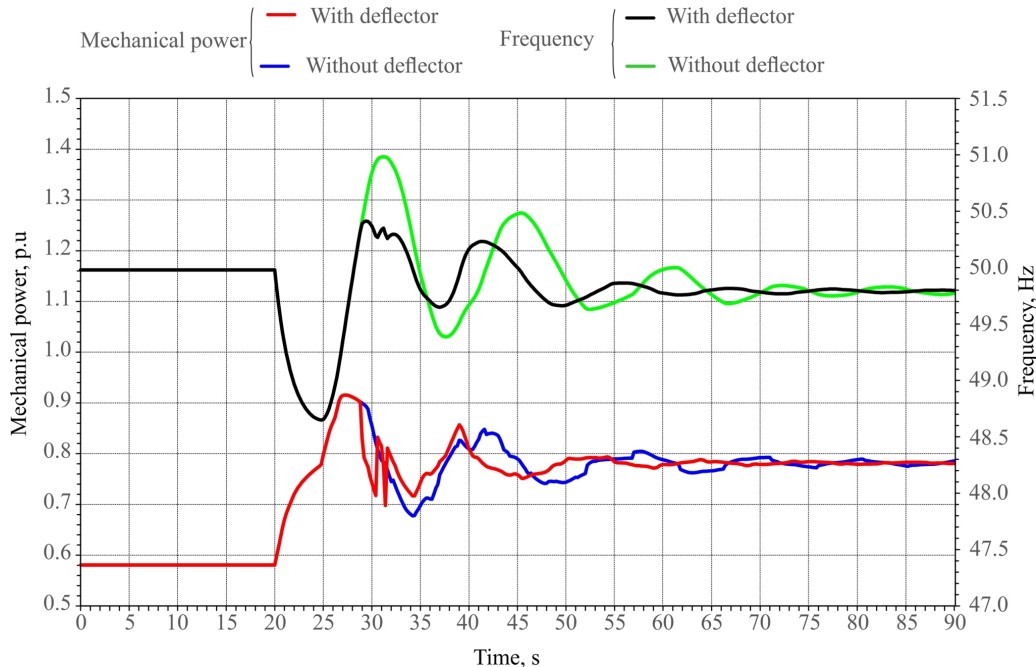

**Figure 22.** Simulation of evolution of mechanical power and frequency with damper and with vs. without deflector.

### 4.3. Validation of the Model Used in the System Simulations

Taking advantage of the fact that the algorithm presented in this work has been implemented and put into service in the electrical system of El Hierro (through the speed regulators of the Pelton hydro turbines of this system), a comparison was made of actual measured data with the data of a similar simulation in order to validate the proposed model. For this, the real and simulated responses of the system in the face of a step loss of generation of approximately 1.6 MW are represented in Figure 23. During the event, three hydro turbines remain coupled which respond to the generation loss. The figure shows the evolution of the measured and simulated values of the frequency of the electrical system and the pressure in the penstock. A high degree of fit between the real and simulated signals can be observed during the time corresponding to primary regulation actuation (around 100 s after generation loss). More specifically, the real and simulated frequency data recorded during this event present a fit of $R^2 = 0.97$, demonstrating the quality of the model in its replication of the measured results, as 97% of the variation of the measured data can be explained with the model. In the case of real and measured pressure, a fit was obtained with the model of $R^2 = 0.91$. In other words, 91% of the variation of the measured data can be explained with the model.

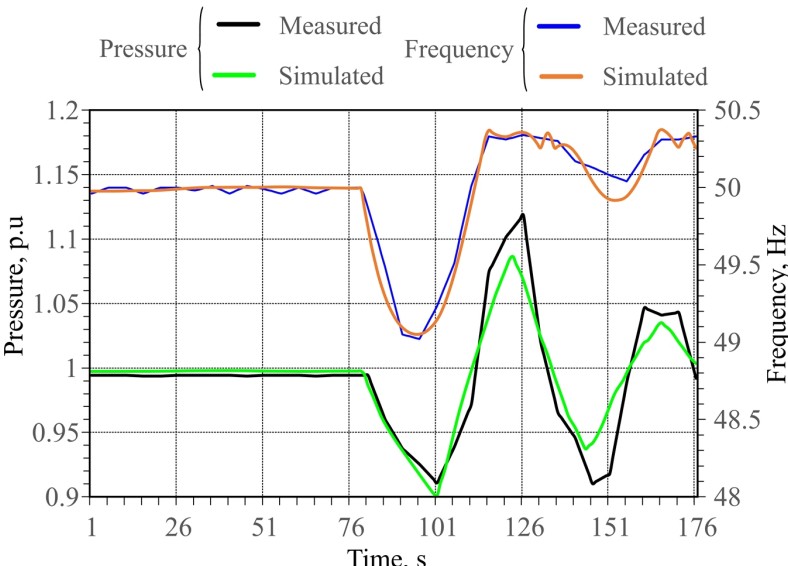

**Figure 23.** Evolution of measured and simulated values of the frequency of the electrical system and conduit pressure in the face of a generation loss of 1.6 MW.

### 4.4. Operation of the Real System before and after Incorporation of the Algorithm

Figures 24 and 25 show the real responses of the electrical system of El Hierro in the face of a similar wind generation loss (of around 1.7 MW), but in two different situations: (i) without incorporation of the damper to improve the control system of the Pelton hydro turbine speed regulators, and (ii) with implementation and incorporation in the control system of the proposed pressure damper. In the first situation (recorded before implementation of the system proposed in this work), a wind generation loss of 1.69 MW took place (Figure 17), and in the second (recorded after the improvement proposed in this work) the generation loss was 1.75 MW (Figure 18). In both cases, it can be seen how the frequency of the system (in black) was affected and its recovery took place within 50 s. However, the behaviour of the different intervening variables and the manner in which this generation loss was corrected differ considerably.

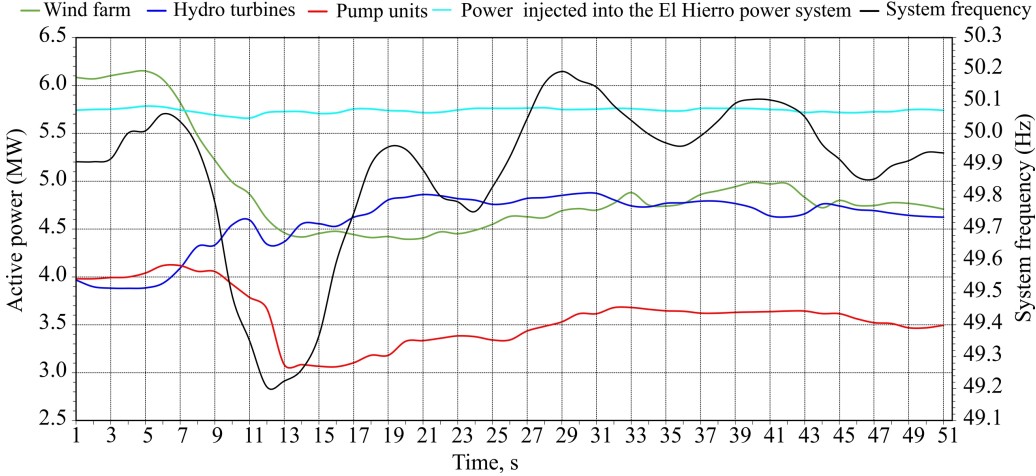

**Figure 24.** Response of the El Hierro electrical system operating without dampers in the face of a generation loss of 1.69 MW.

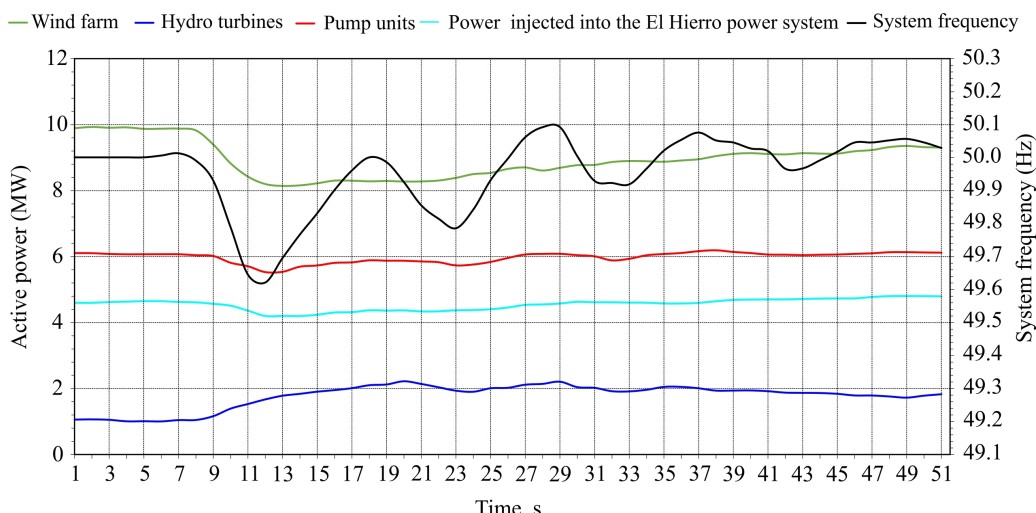

**Figure 25.** Response of the El Hierro electrical system operating with dampers in the face of a wind generation loss of 1.75 MW.

In the first case, the frequency of the system without damper (Figure 24) falls from its initial value of 49.91 Hz to a minimum value of 49.21 Hz, which represents a % frequency loss of 1.41% or a loss of 0.7 Hz. After reaching this minimum value, the frequency recovers because of two interventions. Firstly, the power the system sends to pumping (pump units) is reduced (load shedding), changing from around 4 MW in the initial instant to around 3 MW at its minimum value (red line in Figure 24). At the same time as the power for pumping is reduced, the turbining power of the hydro turbine units (dark blue line in Figure 24) is increased, rising from an initial 4 MW to 4.8 MW at its maximum value.

In the second case (Figure 25), the frequency of the system with damper falls from its initial value of 50.01 Hz to a minimum value of 49.62 Hz, corresponding to % and absolute loss values of 0.76% and 0.39 Hz, respectively. These values are clearly lower than those recorded in the case without damper implementation. Likewise, it can be seen how, after reaching its minimum value, the recovery of frequency differs from its recovery in the case without damper intervention. In this case, the power assigned for pumping (pump units) is only reduced by around 0.4 MW (red line in Figure 25), while the power obtained from the hydro turbines (dark blue line in Figure 25) rises from 1.05 MW to 2.22 MW. That is, in this second case, thanks to damper incorporation, the hydro turbines are able to assume a greater generation loss generating a higher power in a shorter time. In addition, as evidenced in some of the operating results of the El Hierro system (provided by REE) damper implementation has enabled a substantial improvement in the response and stability of the system. Table 2 shows the overall time and the number of times that the frequency of the system exceeded the safety margin of ±250 mHz, as well as the number of times that the safety margin of ±600 mHz was exceeded [58].

**Table 2.** Number of times and overall time that system frequency exceeded different safety margins operating with and without damper over a one year period.

| Damper Available | Analysis Period | Overall Time over ±250 mHz (hours) | No. of Times That Frequency Exceeded the Safety Margin ±600 mHz (#) |
|---|---|---|---|
| No | December 2016–September 2017 | 72.27 | 747 |
| Yes | December 2017–September 2018 | 50.96 | 143 |

Prior to damper installation, the frequency exceeded the safety margin of ±250 mHz for a cumulative time of 72.27 h (annual measure between December 2016 and September 2017). However, after its incorporation, this value fell to 50.96 h (annual measure between December 2017 and September 2018). In addition, the system reduced by 80.9% the number

of times that frequency exceeded the safety margin of $\pm600$ mHz, from 747 to 143 times. Moreover, damper incorporation reduced the number of under-frequency pump unit load shedding events by 93% [58].

## 5. Conclusions

A new algorithm is proposed for the control of the speed regulators of Pelton wheel turbines, used in many of the pumped hydroelectric energy storage systems that operate in isolated electrical systems with high renewable energy participation.

The proposed control system differs substantially from the standard developments which use PID or PI governors in that, in addition to acting on the nozzle needles and deflectors, it incorporates a new inner-loop pressure stabilization circuit to improve frequency regulation and dampen the effects of the pressure waves that are generated when regulating needle position.

The proposed algorithm has been implemented in the Gorona del Viento wind–hydro power plant, replacing the previous control system based on a classical PI governor, and has been validated. The new control system has enabled optimization of the response speed of the turbines in the face of variations in non-dispatchable renewable generation when the frequency control is provided exclusively by the pumped hydroelectric energy storage system, a situation which occurred with a high degree of relative frequency in the case study. The damper has enabled a substantial reduction in the cumulative time and the number of times that frequency exceeded different safety margins. Damper incorporation also reduced the number of under-frequency pump unit load shedding events by 93%.

The design can be implemented in existing or new plants which have a similar topology and which require a response speed increase. The only requirement is to carry out modifications to the speed regulator software of the groups in order to take advantage of the basic control elements of a Pelton turbine.

**Author Contributions:** A.M.: Conceptualization, Software, Methodology, Investigation, Data Curation, Writing—Original draft preparation. J.G.: Investigation, Writing—Original draft preparation. J.A.C.: Formal analysis, Investigation, Methodology, Data Curation, Writing—Original draft preparation, Writing—Review & Editing. Visualization, Supervision, Project administration, Funding acquisition. P.C.: Formal analysis, Investigation, Methodology, Data Curation, Writing—Original draft preparation, Writing—Review & Editing. All authors have read and agreed to the published version of the manuscript.

**Funding:** This research has been co-funded by the ERDF through the INTERREG MAC 2014–2020 programme, within the ACLIEMAC project (MAC2/3.5b/380).

**Institutional Review Board Statement:** Not applicable.

**Informed Consent Statement:** Not applicable.

**Data Availability Statement:** Not applicable.

**Acknowledgments:** The development of this work and the improvement obtained has been possible thanks to, among others, the successful collaboration between REE and Gorona del Viento in the analysis of plant operation and the identification of actions to improve its integration in the El Hierro system. Likewise, we would like to thank the companies Endesa, Instituto Tecnológico de Canarias and Andritz e Idom for their participation during the implementation of the solution in the system. No funding sources had any influence on study design, collection, analysis, or interpretation of data, manuscript preparation, or the decision to submit for publication.

**Conflicts of Interest:** The authors declare no conflict of interest. The funders had no role in the design of the study; in the collection, analyses, or interpretation of data; in the writing of the manuscript; or in the decision to publish the results.

## Nomenclature

**Abbreviation**

| | |
|---|---|
| AGC | automatic generation control |
| DCP | double complex pole |
| ERDF | European Regional Development Fund |
| FDR | fixed damping ratio |
| MPC | model predictive control |
| NE | north-east |
| PHES | pumped hydroelectric energy storage |
| PI | proportional-integral |
| PID | proportional-integral-derivative |
| REE | Spanish initials of the TSO in Spain: Red Eléctrica de España, S.A.U. |
| TSO | transmission system operator |
| WF | wind farm |

**Symbols**

| | |
|---|---|
| $An_i$ | needle opening (function of the position $pn_i$) |
| $A_t$ | turbine gain |
| $a_w$ | wave velocity in [m/s] |
| $d$ | internal conduit diameter [m] |
| $D_{net}$ | single damping constant |
| $D_w$ | rotor speed deviation |
| $Dwi$ | difference between $w_{ref}$ and $w$ |
| $E$ | Young's modulus of elasticity of pipe material |
| $e_n$ | error that regulates needle position |
| $e_p$ | pressure error signal |
| $e_v$ | speed error signal |
| $f$ | thickness of pipe wall [m] |
| $f_p$ | loss factor |
| $f_{pi}$ | friction head loss coefficient in branch pipe |
| $g$ | acceleration due to gravity [m$^2$/s] |
| $GRP$ | generator base MVA for the per unit calculation |
| $h$ | manifold head |
| $H$ | inertia constant |
| $H_b$ | gross pressure head |
| $H_b$ | base head of the penstock |
| $h_e$ | pressure of wave moving along penstock |
| $h_i$ | turbine input pressure |
| $h_l$ | loss of pressure in penstock |
| $h_{li}$ | head loss due to friction in branch pipe $i$ |
| $h_o$ | static pressure (defined by the gross head) of the water column |
| $H_p$ | inertia constant for the needle governor |
| $h_r$ | head per unit in the turbine at nominal flow |
| $k$ | bulk modulus of compression of water [kg/(m s$^2$)] |
| $k_e$ | constant to weigh the difference between instantaneous pressure and $p_{ref}$ in Figure 8 |
| $K_I$ | integral constant of a PI regulator |
| $K_0$ | constant of the feedback in deflector control loop |
| $K_p$ | proportional constant of a PI regulator |
| $L$ | length of the penstock |
| $m$ | needle/deflector curve parameter |
| $M_{eq}$ | sum of the inertia constants of all the hydroelectric units |
| $P_{damping}$ | power from the damping effect due to friction |
| $pd_i$ | deflector positioning |
| $P_{HT}$ | power supplied by the hydroelectric units |
| $P_L$ | power consumed by the external loads |
| $P_{loss}$ | power losses of the turbine |
| $Pm_i$ | mechanical power of turbine $i$ |
| $pn_i$ | position of needle $i$ |
| $P_p$ | power consumed by the pumping system |



| | |
|---|---|
| $p_{ref}$ | pressure that is taken as reference in governors |
| $P_{ti}$ | input power to turbine $i$ |
| $P_{WF}$ | power provided by the non-dispatchable WF |
| $Pw_i$ | hydraulic power of the needle jets of turbine $i$ |
| $Q$ | circulating flow |
| $Q_b$ | flow under the base head |
| $q_i$ | flow which enters penstock $i$ ($i$th penstock) of the manifold |
| $q_{nl}$ | flow without head or the minimum flow necessary for the turbine to provide useful power |
| $q_r$ | flow per unit at nominal head |
| $R_i$ | gain of each hydraulic generation unit |
| $T_1$ | predetermined bound value (Figure 8) |
| $T_e$ | wave travel time in the penstock |
| $T_o$ | predetermined bound value (Figure 8) |
| $T_r$ | time considered to calculate the mean of the $h_i$ generated |
| $TRP$ | turbine MW rating for the per unit calculation |
| $T_w$ | water time constant in the penstock |
| $Tw_l,$ | start times of the water in the branch pipes |
| $w$ | angular speed |
| $w_{ref}$ | reference angular speed |
| $Z_o$ | impedance of the penstock |
| $\rho$ | density of water [kg/m$^3$] |

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
