# Peer review of "A New Control Algorithm to Increase the Stability of Wind–Hydro Power Plants in Isolated Systems: El Hierro as a Case Study"

_jmse, doi:10.3390/jmse11020335_

Round 1

Reviewer 1 Report

Comments and Suggestions for Authors

The paper proposed an algorithm to control the speed of the Pelton wheel turbine in an isolated electrical system with high renewable energy participation. The experimental data validated the simulation results, proving that the algorithm is effective. However, some problems need to be solved in this paper before accepting.

(1)    The creativity of the algorithm should be emphasized in the abstract and conclusions.

(2)    In Figure 14-17, what do the blue cycles represent? In some months, there are two cycles which confuse the readers a lot.

(3)    Please propose how to solve the differential equations in section 2?

(4)    Please check the Nomenclature, such as the ke.

Author Response

Reviewer #1

The paper proposed an algorithm to control the speed of the Pelton wheel turbine in an isolated electrical system with high renewable energy participation. The experimental data validated the simulation results, proving that the algorithm is effective. However, some problems need to be solved in this paper before accepting.

Authors: We would like to thank Reviewer#1 for the opinion given about our manuscript and the conclusions drawn from the results. We also express our gratitude to Reviewer#1 for the time and effort made in the revision of our manuscript and for the recommendations made to improve the quality of our work and which have been implemented in the revised version of our manuscript. Please find below a detailed reply to the comments of Reviewer#1, which we hope we have interpreted correctly.

Reviewer #1: (1)    The creativity of the algorithm should be emphasized in the abstract and conclusions.

Authors: Thank you for this recommendation which we completely agree with. Accordingly, we have modified the title of the manuscript and, following the suggestion of Reviewer|#1, in the Abstract and Conclusions section we have highlighted the creativity of the proposed algorithm implemented in the Gorona del Viento wind-hydro power plant.

Reviewer #1:  (2)    In Figure 14-17, what do the blue cycles represent? In some months, there are two cycles which confuse the readers a lot.

Authors: In Figs. 14-17 different variables are represented using box and whisker plots or diagrams (otherwise known as boxplots). This type of diagram shows a a summary of a large amount of data in five descriptive measures. As well as facilitating intuition of their morphology and symmetry, it also shows any outliers (the blue circles mentioned by Reviewer#1). It is a useful way to compare different sets of data as you can draw more than one boxplot per graph. These can be displayed alongside a number line, horizontally or vertically.

However, we committed the error of not including figure legends to explain the meaning of these boxplots as we had done in other recently published papers [1].

[1] José A. Carta, Pedro Cabrera, Optimal sizing of stand-alone wind-powered seawater reverse osmosis plants without use of massive energy storage, Applied Energy, Volume 304,2021,117888, ISSN 0306-2619, https://doi.org/10.1016/j.apenergy.2021.117888.

We thank Reviewer#1 for pointing out this erratum, which has now been resolved in the revised version of the manuscript. We have now included in each figure a legend which facilitates interpretation of the figure, explaining what the different lines and circles mean on a box and whisker diagram. We have also included a reference [2] if the reader wishes to find out more about such diagrams.

[2] William Navidi. Statistics for Engineers and Scientists. ‎ Sixth edition. McGraw Hill,2023.

Boxplots allow the identification of outliers, represented by the circles mentioned by Reviewer#1. These representations also allow to show the following values:

  • First quartile (Q1):  25% of the values are lower than or equal to this value.
  • Median or second quartile (Q2): Divides the distribution into two equal parts, such that 50% of the values are lower than or equal to this value.
  • Third quartile (Q3):  75% of the values are lower than or equal to this value.
  • Interquartile range (IQR): Difference between the value of the third quartile (Q3) and the first quartile (Q1).

There is no rigorous mathematical definition for what exactly is or is not an outlier, however there are a few tests and criterions that can be applied. These include Chauvernet's criterion, Peirce's criterion, Grubb's test for outliers and Dixon's Q-test. Commonly used rules for recognising outliers include:

lower outlier(s)<Q1-(1.5xIQR)

upper outlier(s)>Q3+(1.5xIQR)

That is to say, the outliers (the circles mentioned by Reviewer#1) are those points which are beyond the lower or upper limits indicated by the edges of the whiskers. An outlier is an unusually large or an unusually small value compared to the others. The outliers found in the aforementioned figures of the manuscript indicate a peculiarity of the process under study.

Reviewer #1:  (3)    Please propose how to solve the differential equations in section 2?

Authors:  We would like to thank Reviewer#1 for this comment as, although in section 4.2, Results of the simulation, we had indicated that “ The deflector and its control were modelled in MATLAB-Simulink following the procedure described in section 2.2.”, we had not specified the tool used to resolve the differential equations of section 2. Thus, following this suggestion of Reviewer#1, we have now indicated the tool used to solve the differential equations. WE consider that the inclusion of this information improves the quality of the manuscript.

Reviewer #1:  (4)    Please check the Nomenclature, such as the ke.

Authors: Following this recommendation, we have checked the Nomenclature, ensuring that ke, for example, has been properly defined.

Reviewer 2 Report

Comments and Suggestions for Authors

The submitted manuscript presents a study of a new management strategy for regulating the speed of a Pelton hydrokinetic  turbine leading to optimization of the response speed in the face of variations in renewable energy generation from a wind-hydro hybrid power plant. The method was applied to Gorona del Viento wind-hydro power plant which fulfils the primary needs of the island of El Hierro. From the obtained results the authors report that in the face of generational changes, the suggested algorithm's implementation is capable of attenuating the pressure wave that rises in the plant's long penstock. As a result, the authors claim there is a reduction in the load shedding that would be necessary to establish stability against the wind farm's disturbances.

The manuscript provides detailed and indepth information, however, following changes are recommended prior to publication:

Major Changes:
1. Please revise t
he abstract to better reflect the significance of the proposed work. Try adding quantitative results to the abstract so the reader has a better preview of the coming attractions to be found in the remainder of the manuscript.

2. The literature review, while detailed, falls short of encompassing sufficient background on wind based renewable energy. The focus seems to be more hydrokinetic turbines, their control strategy and challenges therein. While Wind and Hydro are linked in this study through a hybrid system, wind energy being more stochastic has its own set of challenges. Please try to integrated in your review challenges associated with exploitation of wind energy, particularly in diverse regions of the world.

For example:

1. Assessment of wind-to-hydrogen (Wind-H2) generation prospects in the Sultanate of Oman https://doi.org/10.1016/j.renene.2022.09.116 
2. Forecasting Hydrogen Production from Wind Energy in a Suburban Environment Using Machine Learning https://doi.org/10.3390/en15238901
3. Optimum daily operation of a wind-hydro hybrid system https://doi.org/10.1016/j.est.2022.104540

3. Figure 2 and Figure 3 have a lot of information which is not communicated in the manuscript. Please label the different portions of the figure and try to explain what is happening or alternatively provide simpler figures to substantiate the point you are making in the text. 

4.  Elaborated details such as model/algorithm utilized, software etc. of the simulations are lacking. Please update the case study section with more information for reproducibility of results.

5. In the results section, can you provide a comparison between the outputs of pervious algorithm based solely on a classical PI governor and the proposed algorithm?

Minor Changes

Page 1 Line 11, Heading (Abstract) and paragraph should be separated.

Page 4 Line 144, The bullet point should be corrected to standard format for heading.

Page 4 Section 2, there are multiple paragraphs containing only one sentence. Are some of these headings? This should be organized and/or formatted.

Page 6, Please reconsider figure 4. It seems a simplistic repetition of description provided in text.

Page 8, Figure 6. What happens in the protocol after Opening of Needles?

Page 10-11 Line 305, Line 306, Line 307. Please check figure references.

Page 14 Line 415. Figure 13 displaying layout referenced in line 415 is contradictory to the text presenting the capacity and dimensions of the system. Please check.

Page 20, figures 19 & 20, please add captions to the figures (along with images) for greater clarity.

Overall, the  work is original, novel and important to the field. Formatting of the paper is required along with grammatical review. Please improve the structure and organization of the paper to make the study more concise, understandable and clearer. Multiple paragraphs can be be avoided/merged. Subsequently, the manuscript will be in acceptable format.

Author Response

Reviewer #2

The submitted manuscript presents a study of a new management strategy for regulating the speed of a Pelton hydrokinetic  turbine leading to optimization of the response speed in the face of variations in renewable energy generation from a wind-hydro hybrid power plant. The method was applied to Gorona del Viento wind-hydro power plant which fulfils the primary needs of the island of El Hierro. From the obtained results the authors report that in the face of generational changes, the suggested algorithm's implementation is capable of attenuating the pressure wave that rises in the plant's long penstock. As a result, the authors claim there is a reduction in the load shedding that would be necessary to establish stability against the wind farm's disturbances.

The manuscript provides detailed and indepth information, however, following changes are recommended prior to publication:

Authors: We would like to thank Reviewer#2 for the opinion given about the detailed and in depth information provided in the manuscript. Likewise, we would like to thank Reviewer#2 for the time and effort made in the revision of our manuscript and for the recommendations which have been made to improve the quality of our work.

Reviewer#2:

Major Changes:

  1. Please revise the abstract to better reflect the significance of the proposed work. Try adding quantitative results to the abstract so the reader has a better preview of the coming attractions to be found in the remainder of the manuscript.

Authors: We thank Reviewer#2 for this recommendation and fully concur with the opinion given. We have modified the title of the manuscript and the Abstract in order para to better reflect the significance of the proposed work.  We have added quantitative results to the Abstract so that the reader, as indicated by Reviewer#2, can have a better preview of the contents of the manuscript. We hope to have correctly interpreted the recommendations of Reviewer#2.

Reviewer #2: 2. The literature review, while detailed, falls short of encompassing sufficient background on wind based renewable energy. The focus seems to be more hydrokinetic turbines, their control strategy and challenges therein. While Wind and Hydro are linked in this study through a hybrid system, wind energy being more stochastic has its own set of challenges. Please try to integrated in your review challenges associated with exploitation of wind energy, particularly in diverse regions of the world.

For example:

  1. Assessment of wind-to-hydrogen (Wind-H2) generation prospects in the Sultanate of Oman https://doi.org/10.1016/j.renene.2022.09.116
  2. Forecasting Hydrogen Production from Wind Energy in a Suburban Environment Using Machine Learning https://doi.org/10.3390/en15238901
  3. Optimum daily operation of a wind-hydro hybrid system https://doi.org/10.1016/j.est.2022.104540

Authors: Following the recommendations of Reviewer#2, we have included in the Introduction section aspects associated to the exploitation of wind and solar energy. We have incorporated various references on the exploitation of wind energy con PHES in diverse regions of the world, including the three indicated by Reviewer#2 as examples.

Reviewer #2: 3. Figure 2 and Figure 3 have a lot of information which is not communicated in the manuscript. Please label the different portions of the figure and try to explain what is happening or alternatively provide simpler figures to substantiate the point you are making in the text. 

Authors: Following the suggestion of Reviewer#2, we have now included in the text more information about the content of the figures.

Reviewer #2: 4.  Elaborated details such as model/algorithm utilized, software etc. of the simulations are lacking. Please update the case study section with more information for reproducibility of results.

Authors: We would like to thank Reviewer#2 for the interest in improving the quality of our manuscript. In this regard, we have added new text in each subsection of the tasks described in section 2, indicating the specific functions that we used to implement each block diagram described in Matlab-Simulink.

Reviewer #2: 5. In the results section, can you provide a comparison between the outputs of pervious algorithm based solely on a classical PI governor and the proposed algorithm?

Authors:  Again, we would like to thank Reviewer#2 for this suggestion and point out that this comparison of results is available in Table 2 of the manuscript.

Reviewer #2:

Minor Changes

Page 1 Line 11, Heading (Abstract) and paragraph should be separated.

Page 4 Line 144, The bullet point should be corrected to standard format for heading.

Page 4 Section 2, there are multiple paragraphs containing only one sentence. Are some of these headings? This should be organized and/or formatted.

Authors: The authors would like to express our thanks for these detailed comments of Reviewer#2. We concur with the need to resolve the formatting problems, but would like to point out that the formatting errors detected were due to a rapid editing prepared by the Journal for this review process. All the comments of Reviewer#2 will be taken into account for the final editing of the paper.

Page 6, Please reconsider figure 4. It seems a simplistic repetition of description provided in text.

Authors: We would prefer to maintain Figure 4, as we consider it offers the potential reader greater clarity and a better understanding of the descriptions provided in the text.

Page 8, Figure 6. What happens in the protocol after Opening of Needles?

Authors: We thank Reviewer#2 for this comment as, thanks to the comment, we have detected an erratum in Figure 6. This has been corrected and what happens after needle opening is now known.

Page 10-11 Line 305, Line 306, Line 307. Please check figure references.

Authors: We thank Reviewer#2 for this comment as, thanks to the comment, we have detected an erratum in the reference to Figure 8. This has now been corrected.

Page 14 Line 415. Figure 13 displaying layout referenced in line 415 is contradictory to the text presenting the capacity and dimensions of the system. Please check.

Authors: We thank Reviewer#2 for this comment and have moved the reference to another part of the text e trust that we have interpreted correctly this comment of Reviewer#2.

Page 20, figures 19 & 20, please add captions to the figures (along with images) for greater clarity.

Authors: We thank Reviewer#2 for this suggestion, which increases the clarity and understanding of Figures 19 and 20, which have now been modified in the revised version of the manuscript.

Reviewer#2: Overall, the  work is original, novel and important to the field. Formatting of the paper is required along with grammatical review. Please improve the structure and organization of the paper to make the study more concise, understandable and clearer. Multiple paragraphs can be be avoided/merged. Subsequently, the manuscript will be in acceptable format.

Authors: Firstly, we would like to thank Reviewer#2 for the favourable opinion given about the originality, novelty and importance of our work to the field. With respect to the comment of Reviewer#2 that “Formatting of the paper is required along with grammatical review”, we would like to state the following. Before submitting this particular manuscript for possible publication in JMSE, it was revised and checked for possible spelling and/or grammar mistakes by a professional linguist and native English speaker. However, given that Reviewer#2 has not indicated precisely which language problems the paper suffers from and since, despite our best efforts, we have been unable to detect exactly which problems are being referred to, we have had the manuscript checked again, this time by another native English speaker and graduate in Literae Humaniores at Oxford University who has worked as a professional linguist for the past 30 years. We received the following answer:

“As requested by the authors, I have read through their manuscript to resolve any language problems. I have made a few, very minor, modifications to some sentences to enhance their fluidity, but otherwise found the text perfectly understandable and without any grammar mistakes.”

As for the structural organization of the work, we would like to inform Reviewer#2 that we have now included various subsections in section 2. We trust that we have correctly interpreted the suggestions made by Reviewer#2.

Reviewer 3 Report

Comments and Suggestions for Authors

1. The main contribution of the paper is new inner-loop pressure stabilization circuit to improve frequency regulation and dampen the effects of the pressure waves. However, neither the abstract nor the title of the paper give a hint on what type of strategy was implemented. Since this proposed speed regulation is the main contribution of the paper, it should be given more prominence.

2. Reference [19] (cited in lines 59-61) is from 2007. Are the problems related to turbine regulation with large load variations in the electrical system yet unresolved in 2023 and continue to pose challenges to the control community? Hydroelectric power is a very mature technology. Do you have some more recent references demonstrating this is still an issue?

3. Subsections of section 2 "Modeling of the system" are bad formatted. Shouldn't "General configuration...", "Basic outline..." and "Tasks covered by the proposed..." all be numbered subsections?

4. Please, improve the quality of Fig. 3, some variables indexes are difficult to read.

5. Subsection 2.5 should be "Task 5" and subsection 2.6 should be "Task 6".

6. Why there is no subsection describing "Task 7 - Automatic generation control modeling"?

7. In line 498, shouldn't the time instant be  t=32.6 s?

8. In table 2, are the frequency safety margins in MHz? Shouldn't it be mHz ?

Author Response

Reviewer #3

Reviewer #3: 1. The main contribution of the paper is new inner-loop pressure stabilization circuit to improve frequency regulation and dampen the effects of the pressure waves. However, neither the abstract nor the title of the paper give a hint on what type of strategy was implemented. Since this proposed speed regulation is the main contribution of the paper, it should be given more prominence.

Authors: We would like to thank Reviewer#3 for indicating the issues that we need to address to improve the quality of the work we have undertaken. Following the recommendation of Reviewer#3, we have modified the title and the content of the Abstract in the new version of the manuscript. In the Abstract, we have described the proposed control algorithm implemented in the Gorona del Viento wind-hydro power plant. We have tried to give more prominence to the proposed speed regulation algorithm, as suggested by Reviewer#3.  We hope that we have correctly interpreted the suggestions made by Reviewer#3.

Reviewer #3: 2. Reference [19] (cited in lines 59-61) is from 2007. Are the problems related to turbine regulation with large load variations in the electrical system yet unresolved in 2023 and continue to pose challenges to the control community? Hydroelectric power is a very mature technology. Do you have some more recent references demonstrating this is still an issue?

Authors: As indicated in the Introduction section, and as indicated by Reviewer#3, hydroelectric power is a very mature technology. However, control systems implemented in PHES with wind energy that aim to have the frequency control provided exclusively by the PHES system continue to be a challenge in the control community. Indeed, as shown in section 1.1 “Literature review on control to increase wind-hydro power plant stability in isolated systems”, while various algorithms have been proposed they have not been validated. The authors of recently published studies stress that the controllers should be implemented and properly tested in a laboratory to better verify their performance.

Reviewer #3: 3. Subsections of section 2 "Modeling of the system" are bad formatted. Shouldn't "General configuration...", "Basic outline..." and "Tasks covered by the proposed..." all be numbered subsections?

Authors: Following the recommendations of Reviewer#3, we have now incorporated the suggested subsections in the new version of the manuscript.

Reviewer #3: 4. Please, improve the quality of Fig. 3, some variables indexes are difficult to read.

Authors: We thank Reviewer#3 for pointing out this difficulty in reading the indices of some of the variables contained in Fig.3. We have increased their size in the new version of the manuscript to make them easier to read.

Reviewer #3: 5. Subsection 2.5 should be "Task 5" and subsection 2.6 should be "Task 6".

Authors: We thank Reviewer#3 for pointing out these errata in the titles of subsections 2.5 and 2.6. (2.3.5 and 2.3.6 in the new version of the manuscript). These errata have now been corrected.

Reviewer #3: 6. Why there is no subsection describing "Task 7 - Automatic generation control modeling"?

Authors: We would  like to thank Reviewer#3 for this comment. We considered it unnecessary to describe the Automatic Generation Control model as it is widely known and used in various references. Nonetheless, in this respect we have now added the following description immediately before subsection 2.3.1: “In task 7, as is widely known, estimation is made of the powers generated by each hydraulic generation unit according to their gains Ri. That is, the amount of load pickup on each unit is proportional to the slope of its droop characteristic [48].”

Reviewer #3: 7. In line 498, shouldn't the time instant be  t=32.6 s?

Authors: We would like to thank Reviewer#3 for pointing out this erratum, which has now been corrected in the new version of the manuscript.

Reviewer #3: 8. In table 2, are the frequency safety margins in MHz? Shouldn't it be mHz ?

Authors:  We would like to thank Reviewer#3 for pointing out this erratum, which has now been corrected in the new version of the manuscript.

Round 2

Reviewer 1 Report

Comments and Suggestions for Authors

I agree to publish this paper. Thanks. 

Reviewer 2 Report

Comments and Suggestions for Authors

The authors have made appropriate changes to the manuscript in the revised version to address the queries/concerns raised by this reviewer.